# Active subglacial lakes and channelized water flow beneath the Kamb Ice Stream

Byeong-Hoon Kim[1], Choon-Ki Lee[2*], Ki-Weon Seo[1], Won Sang Lee[2,3], Ted Scambos[4]

[1]Department of Earth Science Education, Seoul National University, Seoul, 151-742, Republic of Korea
[2]Unit of Ice Sheet and Sea Level Changes, Korea Polar Research Institute, Incheon 21990, Republic of Korea
[3]Polar Sciences, Korea University of Science and Technology, Daejeon 34113, Republic of Korea
[4]National Snow and Ice Data Center, University of Colorado, Boulder, Colorado 80309-0449, USA

*Correspondence to*: Choon-Ki Lee (cklee92@kopri.re.kr)

**Abstract.** We identify two previously unknown subglacial lakes beneath the stagnated trunk of the Kamb Ice Stream (KIS). Rapid fill-drain hydrologic events over several months are inferred from surface height changes measured by CryoSat-2 altimetry and indicate that the lakes are probably connected by a subglacial drainage network. The subglacial drainage network structure is inferred from the regional hydraulic potential, and it probably links the lakes. The sequential fill-drain behavior of the subglacial lakes and concurrent rapid thinning in a channel-like topographic feature near the grounding line implies that the subglacial water repeatedly flows from the region above the trunk to the KIS grounding line and out beneath the Ross Ice Shelf. Ice shelf elevation near the hypothesized outlet is observed to decrease slowly during the study period. Our finding supports a previously published conceptual model of the KIS shutdown stemming from a transition from distributed flow to well-drained channelized flow of sub-glacial water. However, a water-piracy hypothesis in which the KIS subglacial water system is being starved by drainage in adjacent ice streams is also supported by the fact that the degree of KIS trunk subglacial lake activity is relatively weaker than those of the upstream lakes.

## 1 Introduction

The basal hydrology of the Siple Coast ice streams (i.e. the Kamb, Whillans, Binschadler and MacAyeal ice streams) plays a critical role in the ice dynamics of this region and its ongoing evolution (Bell, 2008; van der Wel et al., 2013). The Kamb Ice Stream (KIS), located on the eastern boundary of the Ross Ice Shelf, ceased streaming ice flow approximately 160 years ago (Retzlaff and Bentley, 1993). This event significantly affected the mass balance of the West Antarctic Ice Sheet (WAIS), locally incurring a net mass gain equal to ~20% of the net mass loss of the WAIS (Rignot et al., 2008). Hulbe and Fahnestock (2007) and Catania et al. (2012) have suggested that the Siple Coast ice streams have experienced several stagnation and reactivation cycles. Abrupt change in the basal hydrological system has been cited as a possible cause for the stagnation in several studies (Anandakrishnan and Alley, 1997; Catania et al., 2006; Retzlaff and Bentley, 1993). van der Wel et al. (2013) also showed numerically that the period of the long-term velocity cycles in the KIS is strongly associated with its subglacial hydrology, ice thermodynamics, and till regime. All these factors are related to the basal melt rate and

upstream subglacial water supplies. One hypothesis for the KIS stagnation is that it resulted from a change in the configuration of the subglacial drainage system from sheet flow to channelized water flow (Retzlaff and Bentley, 1993), although no previous direct evidence of channelized flow beneath the KIS has been observed. Another hypothesis suggests that reduced lubrication of the KIS basal interface, caused by a change of the subglacial water pathway in an upstream region, provoked the stagnation of the downstream region. This is known as the water-piracy hypothesis (Anandakrishnan and Alley, 1997).

There are several subglacial lakes (SGLs) underneath the KIS (Figure 1). Although the existence of active SGLs beneath the KIS had been revealed earlier by surface height variation from ICESat laser altimetry (Smith et al., 2009) and RADARSAT radar interferometry (InSAR) (Gray, 2005), little was known about the hydrological connections between adjacent lakes in the KIS. The northern corner of the KIS, part of a region informally called 'the Duckfoot' (see Fig. 1 in Fried et al. (2014)), is thought to contain a marginal lake, due to thinning along the northern shear margin and retreat of the grounding line (Fried et al., 2014); similar to Subglacial Lake Engelhardt downstream of Whillans Ice Stream (WIS) (Fricker et al., 2007; Fried et al., 2014). Simulations of present-day subglacial hydrology also suggest that a subglacial channel may still exist beneath the KIS (Carter and Fricker, 2012; Goeller et al., 2015) and basal channels in the floating ice shelf near the KIS trunk are observed (Alley et al., 2016). However, the ICESat repeat-track method is limited by the sparse spatial and temporal coverage of the ground tracks, making it difficult to determine the lake boundaries in detail, time glacial lake fill-drain cycles, or map small lakes (McMillan et al., 2013). The InSAR method is also hampered by low temporal resolution and coverage. Thus, previous studies have not detected any signal associated with subglacial lake activity in the stagnated trunk of the KIS.

The CryoSat-2 radar altimeter, launched in mid-2010, has provided topographic measurements of the Antarctic Ice Sheet with better spatial resolution than previous radar altimeters, and much better than that of ICESat (Wingham et al., 2006a). In this study, we investigate the elevation changes related to subglacial activity using the CryoSat-2 measurements from 2010 to 2015 and report the presence of two previously unknown subglacial lakes whose behavior is probably related to the activities of known upstream lakes. In addition, we identify the probable influence of subglacial water activity on the grounding line of the KIS, using available ICESat/CryoSat-2 altimetry measurements and Landsat optical images. Our findings lead us to several conclusions regarding the characteristics of the basal hydrologic environment under the KIS.

## 2. Data & Methods

### 2.1 CryoSat-2 data

CryoSat-2 operates in three different modes: Low Resolution Mode (LRM), Synthetic Aperture Radar (SAR), and SAR interferometric (SARin) (Wingham et al., 2006a). The SARin mode coverage includes the margins of the ice sheets and mountainous regions. For the detection of unknown lakes, we primarily utilize the Level 2 (L2) product of the SARin mode, which directly provides geospatially corrected surface elevations with various correction terms and error flags. We use both

baseline B (July 2010 – February 2015) and baseline C (March 2015 – June 2015) products. The -0.67 m CryoSat-2 instrument bias in the baseline B product is removed before processing (McMillan et al., 2013). The SARin mode provides a determination of the precise reflecting (backscattering) point on the surface with nominal spatial resolutions of ~300 m and ~1.5 km for the along- and across-track directions respectively, resulting in the irregular spatial distribution of reflecting

points (Wingham et al., 2006a). Wang et al. (2015) has reported that the vertical accuracy of the SARin mode ranges from 0.17 m to 0.65 m in the interior of ice sheets and on ice shelves, although the magnitude of the surface height error depends on the slope of the imaged surface.

We also utilize the L2 elevation product of the LRM mode, which is measured by a single antenna as in conventional pulse-limited radar altimetry (Wingham et al., 2006a). The geographical mask of the LRM covers the interior

of the ice sheet, especially upstream of lake Kamb Trunk 1 (KT1; Figure 1). The nominal pulse-limited footprint of the LRM is about ~1.65 km in both along- and across-track directions, which is larger than that of the SARin mode. Moreover, because the LRM mode cannot obtain exact backscatter points on the undulating ice surface, it is expected that the LRM elevation data have lower accuracies than the SARin mode. Despite a larger footprint and lower accuracy, we use LRM mode data in a manner similar to our SARin mode method to verify the upstream lake activities.

## 2.2 Subglacial lake detection from CryoSat-2 measurements

To detect SGLs in the study area, we adopt the data processing method used in McMillan et al. (2014). We remove the low-quality returns, with the height error flags indicating errors in height determination, or extremely high backscatter values (> 30 dB), and then recursively apply a 3-sigma filter to the elevation residuals deviated from a recent Cryosat-2

based DEM product (Helm et al., 2014). To estimate an elevation change rate in a 5 × 5 km region, a quadratic curved surface fitting the elevation measurements in the study period (July 2010 to June 2015) is determined and then removed from the elevation measurements. From the topography-free elevation residuals, we estimate the elevation change rate by linear fitting to the data, within a constant time window of two-years, successively shifting the time window by 1-month intervals. This method increases the reliability of the inferred rate changes in successive time windows, and avoids some of the

uncertainty in estimating the duration of a rate change, by avoiding the smoothing effects of a quadratic fit to the overall data pattern (as in McMillan et al. (2014)). To increase the spatial resolution, the 5 × 5 km regions used for the elevation change rate are overlapped by 1km.

Using the resulting successive maps of elevation change rate, we inspect the spatiotemporal variation of the elevation change rate and select the KIS SGL candidate areas. For example, Figure 2 shows the elevation change rate and its

uncertainty (linear regression error in each 5 × 5 km region) in a two-year time window from February 2012 to January 2014 in the trunk of the KIS. We identify two subglacial lakes based on their surface elevation change rates (>1 m/yr) and low uncertainties (~0.3 m/yr) using our analysis. Hereafter, we call the two lakes Kamb Trunk 2 (KT2) and 3 (KT3), because they are located downstream of lake KT1, which was already identified by analysis of ICESat data (Smith et al., 2009).

There are a few additional regions with anomalous elevation change rates besides KT2 and KT3, but the anomalies do not sufficiently exceed their uncertainties (> 0.5 m/yr).

**2.3 Subglacial lake boundary**

In order to specify the boundaries of the SGLs with greater detail, we use digital elevation models (DEMs) generated from CryoSat-2 measurements. We first generate a reference DEM with 100 m resolution from CryoSat-2 elevations during July 2010 – December 2011, using a kriging method (Goovaert, 1997) (Figure 3a). The reference DEM is compared to other DEMs generated for various time windows. The time windows of May 2013 – January 2015 for KT1, and July 2013 – January 2014 for KT2 and KT3, yield the clearest elevation anomalies indicating the SGL boundary (Figure 3b). A contour line with the same value as the standard deviation of the elevation anomalies on nearby stationary ice is empirically chosen as the lake boundary. This is in good agreement with the lake boundary independently inferred from the repeat-track analysis of ICESat measurements for KT1 (see Section 2.5).

**2.4 Elevation and volume changes of subglacial lakes and their uncertainty**

After removing the reference DEM elevations from the CryoSat-2 measurements, the residuals within the lake boundary are averaged in monthly intervals to generate the time series of elevation change. The background elevation time series, estimated using the elevations in a donut-shaped area of 2-km width around each lake, are removed to highlight the elevation changes associated with the SGL's activities. The background elevation changes represent gradual thickening of ice up to ~1 m over the study period. Volume change is calculated by simply multiplying the time series of elevation change and the area of lake determined by the lake boundary. The uncertainty of the elevation change time series is calculated as the errors in elevation on adjacent stationary ice sheet areas, similarly to the method of Wingham et al. (2006b). For our error estimates, we calculate the standard deviation of the residuals as the error of the lake elevation time series, after removing the linear trend of the background elevation time series. The error for the volume change time series is derived from the error of the elevation time series and a 10% error of lake area.

**2.5 ICESat data and repeat-track analysis**

ICESat measures surface elevation with an accuracy of ~14 cm, a footprint size of ~65 m and an along-track interval of 172 m (Shuman et al., 2006). We use the ICESat GLA12 (Release-34) data products acquired from October 2003 to April 2009 (campaigns Laser 2a to Laser 2e). High gain (>200) records are rejected and saturation correction is applied to the L2 product of ICESat (Pritchard et al., 2012). The inter-campaign biases are corrected using the values determined by the data collections close to latitude 86°S (Hofton et al., 2013). In order to remove the influence of surface slope or topography on the estimation of the elevation change (i.e. slope correction), we remove the elevations from the reference DEM mentioned in Section 2.3 from all ICESat measurements. The elevation change rates are estimated along repeat tracks using the residual elevations. The reference ground tracks of ICESat are divided into 172 m intervals generating the points at

which the elevation change rates are estimated. The shots within 300 m of each point are gathered, and the elevation change rates at each point are estimated by linear fittings of the residual elevations at the gathered shots.

## 2.6 Hydraulic potential

Movement of subglacial water is mainly governed by two factors: bedrock topography and overburden ice thickness. The hydraulic potential beneath the ice sheet is calculated as follows:

$$P_h = \rho_w g z_b + \rho_i g z_i$$

where $\rho_w$ and $\rho_i$ are the density of water (1000kg/m$^3$) and ice (917kg/m$^3$), $z_i$ and $z_b$ are ice thickness and bedrock elevation with respect to the geoid, and $g$ is gravitational acceleration (Shreve, 1972). We subtract a constant of 250 kPa to set the

hydraulic potential near the grounding line to 0 kPa. We use the ice thickness and bedrock elevation from BEDMAP2 (Fretwell et al., 2013). The subglacial streamlines are generated from the hydraulic potential gradient using a topographic analysis software, TopoToolbox (Schwanghart and Scherler, 2014).

## 3 Results and Interpretation

### 3.1 Hydrological connectivity of subglacial lakes in the Kamb Ice Stream

We identify two previously unknown SGLs in the trunk of the KIS (Figure 1). The lakes are located in areas characterized by both local surface topographic lows and hydraulic potential lows (Figure 3). The DEM differencing mentioned in Section 2.3 shows clear elevation changes, coinciding with the hydraulic potential hollows. The maxima of the elevation anomalies inside the lakes are in the range of 3 to 5 m. The areas of KT1, KT2, and KT3 are 43.5, 31.7 and 38.7 km$^2$, respectively. Streamlines derived from the hydraulic potential gradient map pass though the lakes. The lakes are also

located on a 'potential subglacial lake' area, identified from previous analysis of continent-wide subglacial hydraulic potential (Livingstone et al., 2013).

   Figure 4 shows the elevation and volume changes of three SGLs, representing sequential filling events in 2013. The error range, indicated by transparent coloring, is empirically determined as the standard deviation of elevation measurements on the stationary ice adjacent to each lake. The volume of lake KT1 begins to increase in early 2013. Roughly two months

later, the volume of lake KT2 starts to increase, and another two months later, the volume of lake KT3 also increases, exceeding the mean volume variations (~0.03 km$^3$) before 2013. The volume of lake KT1 increases by ~0.1 ± 0.03 km$^3$ over ~6 months, which indicates that the filling rates (i.e. the balance of the inflow and outflow rates) is approximately 6 ± 2 m$^3$/s. The sudden volume increases of lakes KT2 and KT3 also indicate similar filling rates (8 ± 2 m$^3$/s for KT2 and 9 ± 2 m$^3$/s for KT3). Sequential drops in lake volume after the filling events are also observed, but in the opposite order. The excess water

in lake KT3 was completely drained in 8 months after the start of the filling event, whereas the lake KT2 returned to its previous level in 16 months. Lake KT1 had not returned to its previous level by the end of the study period. In Figure 4, the

high-amplitude fluctuations in the time series observed when the lakes were filled appears to be an artefact of non-uniform spatial sampling of the elevation anomalies.

It is interesting to note that the volumes of the downstream lakes KT2 and KT3 begin to increase before the upstream lakes are entirely filled. Another important observation is that there is no volume change in lake KT3 during the drainage stage of KT2 in the middle of 2014. These two facts may indicate significant hydrological characteristics of the lake system, as discussed later (see Section 4). As shown in Figure 3c, the hydraulic potentials at the outlets of lakes KT1, KT2, and KT3 (inferred along the streamline) are 30–80 kPa higher than the minima inside the lakes. The potential differences are equivalent to the head differences of 3–8 m, roughly consistent with the maximum elevation changes inside the lakes (3–5 m) when the lakes are fully filled. Therefore, to account for the early filling of the downstream lakes, the lake water must flow over these hydraulic barriers before the hydraulic head reaches the full capacity of the lake.

According to the model study of Goeller et al. (2015), the three SGLs in the trunk of the KIS appear to be connected with SGLs in the region upstream of the KIS trunk where the ice stream is still flowing. Unfortunately, the geographical mask of the CryoSat-2 SARin mode does not cover the upstream KIS area, but the LRM mode is available. Using the LRM mode products, we detected large elevation changes in three lakes, Kamb 1 (K1), Kamb 3 (K3), Kamb 4 (K4), and Kamb 8 (K8), upstream of the KIS. Other upstream lakes in the KIS do not show any apparent elevation changes during the study period. All the lakes have been reported by Smith et al. (2009), but here we rename the K3 and K4 as K34 because they seem to be a single lake (Figure 5a). In early 2012, the surface elevation of lake K34 begins to decline, implying a water discharge event following the rapid filling in late 2011. At the peak of lake K34's volume (January 2012), K1 begins to increase in volume, strongly suggesting a linkage between the upper (K34) and lower (K1) lakes (Figure 5c). The discharge of lake K1 begins in January 2013 and continues until June 2015. The timing of the discharge from the lake K1 is coincident with, or slightly precedes, a filling event of KT1 (February 2013), implying that the lake K1 is supplying the water to the lakes in the KIS trunk. On the other hand, further upstream of the KIS, lake K8 shows a sudden elevation loss in late 2012, suggesting a water drainage event. However, any connection between K8 drainage and other SGLs' activities in the KIS is not clear, since a pathway from lake K8 to lake K1 or the KIS trunk lakes cannot be clearly identified (Goeller et al., 2015).

**3.2 Influence of subglacial water on the Kamb Trunk estuary**

The topographically low area downstream of KT3 is interpreted here as a kind of subglacial 'estuary'. The hydraulic potential shows a broad area of low values in this region. A slight lowring of surface elevation (i.e. thinning of the ice) is seen over the estuary area, using both ICESat (from 2003 to 2009) and CryoSat-2 measurements (from 2010 to 2015; Figure 6b and Figure 6c). In order to combine the time series of elevation averaged over the estuary region from ICESat and Cryosat-2, we correct the bias (~1 m) between ICESat and Cryosat-2 elevations due to radar penetration into the snow pack (Davis and Moore, 1993), which is estimated by the comparison of both elevation measurements along an adjacent ICESat track on stationary ice. The combined time series show a persistent elevation lowering of ~0.12 m/yr on average during the study period.

An ice surface feature probably indicating channelized subglacial flow in the estuary region is observed in satellite imagery, as roughly indicated in Alley et al. (2016). The background MOA image in the blue rectangle of Figure 6a shows a narrow sink near the grounding line downstream of KT3. A detailed examination of Landsat images over the last two decades shows that the feature is continuously extending upstream (Figure 7). Considering its extending rate (~100 m/yr), the length from the grounding line (8–9 km) and the retreat rate of the grounding line (~30 m/yr from Thomas et al. (1988)), the channel-like feature may have begun to form around 110 years ago, i.e., not long after the stagnation of the KIS. The feature is too concave for CryoSat-2 to measure its inside elevations, because the radar signals reflected from the rim around the sink arrive in advance. However, ICESat laser returns from within the channel-like feature show a large elevation decrease (~1.2 ± 0.1 m/yr average using a linear fit along ICESat track 221; see Figure 7). We posit that this feature was formed by a basal melting induced by the outflow of subglacial water into the sub-ice-shelf cavity as discussed in the next section.

## 4 Discussion

Wingham et al. (2006b) reported that three inferred subglacial lakes along a ~100 km line in the Adventure Trench Region experienced near-simultaneous filling by the water supply from an upstream lake. To explain this behavior, Carter et al. (2009) suggested that the high variability of local water pressure in (hypothesized) turbulent water flow is more significant than a few meters of hydraulic head difference as a barrier between adjacent lakes. Similarly, Fricker and Scambos (2009) have observed a linked drainage event between Lake Conway and Lake Mercer in the Whillans/Mercer Ice Stream, but the volume changes of the two lakes were not always explained by a direct relationship between the drainage from upstream and filling of downstream reservoirs. They explained that the opening of an outlet conduit for the downstream lake allows the additional floodwater from the upstream lake to move directly through the downstream lake without increasing its water level. Similarly, the early filling of downstream lakes (KT2 and KT3) before the upstream lakes (KT1 and KT2) are fully filled implies that the three lakes in the trunk of KIS are plausibly connected by conduits that may be easily opened by turbulent water flow. The water that is drained from KT2 and passed directly through KT3 in the middle of 2014 also suggests the opening or growth of a conduit caused by flood events. If this mechanism is operating, it diminishes the role of low hydraulic barriers. In comparison to the behaviors of K1 and K34, which show connectivity more typical of subglacial lakes (i.e. draining of the upstream lake and filling of the downstream lake), we suppose the lakes in the KIS trunk are controlled by more complex mechanisms in the subglacial lake-channel system.

If we assume the filling rate of lake KT2 ($8 \pm 2$ m$^3$/s) is equal to the discharge rate ($Q$) from lake KT1, a semi-circular tunnel with a cross-section ($S$) of $5 \pm 1$ m$^2$ could support the discharge with an average hydraulic potential gradient of ~10 Pa/m between KT1 and KT2, according to "R-channel" theory (Röthlisberger, 1972). Based on the same calculations of R-channel basal hydraulics, as described in the supplementary method of Wingham et al. (2006b), the total energy released by the flow between KT1 and KT2 is mostly consumed by melting the tunnel roof rather than heating the water and

deforming the roof of lake KT2. Therefore, the melt rate is approximated as $Q\Delta\Phi/Ll\rho_i = 2.8 \times 10^{-7}$ m$^2$/s, where $Q$ (8 m$^3$/s) is the discharge rate, $\Delta\Phi$ (=1.8 kPa) is the hydraulic potential difference between KT1 and KT2, $l$ (= 160 km) is the distance between KT1 and KT1, $L$ (= $3.3 \times 10^5$ J/kg) is latent heat of water, and $\rho_i$ (= 917 kg/m$^3$) is density of ice. The creep closure rate of the tunnel is given by $ASP_e^n/(n\text{-}1)2^n$ , where $A$ and $n$ are flow parameters, $S$ is cross-section of tunnel, and $P_e$ is

effective pressure (Nye, 1953). Using $A = 2.5 \times 10^{-24}$ Pa$^{-3}$/s at the melting point, $n = 3$, and $S = 5$ m$^2$, the effective pressure required to balance the growth by melting and the creep close of tunnel is ~700 kPa. However, the increase in reservoir pressure at lake KT2 ($\rho_i g\Delta h$), where $g$ is gravitational acceleration and $\Delta h$ (= 1.7 m) is the elevation change of lake KT2, is only ~17 kPa, which is much smaller than the effective pressure required for the creep close of tunnel (~700 kPa). These simple calculations, similar to those of Wingham et al. (2006b), roughly verify that a conduit between KT1 and KT2 can be

supported by the melting due to the discharge of subglacial water. However, Alley et al. (1998) have suggested that an R-channel model may not be applicable beneath an ice sheet with an adverse bed slope, as the KIS. Fowler (2009) also suggested that channels in the underlying sediment (Weertman, 1972) might be the usual mechanism in the Antarctica. More recently, Carter et al. (2016) numerically reproduced the observed lake volume changes in the Whillans/Mercer Ice Streams using a basal water model including a single channel incised into the subglacial sediment. Considering their modeling result

for Subglacial Lake Whillans (SLW), which has a lake area (58 km$^2$), inflow rate (4 m$^2$/s) and bed topography pattern similar to those of the SGL system in the KIS trunk, we can infer that a canal flow could exist recurrently in the KIS trunk too.

Considering the thinning of ice in the KIS trunk estuary (Figure 6), we presume that a small part of the basal water flow from KT3 is converted to distributed (sheet) basal flow in the estuary, since the hydraulic potential around the estuary has a fan-shaped pattern (Figure 6a) and ice-penetrating radar data near the outlet of lake KT3 shows a very flat bed

topography (Catania et al., 2006). If the distributed basal flow exists alongside channelized flow, it increases the lubrication of the ice sheet bed and decreases the drag force on the ice sheet flow. The basal lubrication could enhance the longitudinal stress due to the tensile forces inherent between the moving ice shelf and the stagnant grounded ice and drive the thinning of the ice sheet in the lubricated estuary region. This explanation is supported by the fact that the ice sheet around the estuary is not entirely stagnant, based on velocity mapping from InSAR (Rignot et al., 2011a).

The KIS trunk estuary shows some similarities to the WIS trunk estuary reported by Horgan et al. (2013) and Christianson et al. (2013), i.e. similarities in the shapes of the estuaries, potential saddles dividing the estuaries from upstream potential lows and upstream subglacial lakes which probably supply subglacial water into the estuaries. However, the KIS trunk estuary is located on the upstream side of the current groundling line and has higher hydraulic potentials (300 – 500 kPa), indicating that it is entirely grounded at present; whereas the estuary downstream of the WIS, with almost 0 kPa

potential, is probably exchanging water and sediment across the grounding zone through viscoelastic flexure induced by tidal forcing. If the current thinning of the ice sheet over the KIS trunk estuary (~0.1 m/yr) are maintained in the future, the ice would begin to partially float after a few centuries. Horgan et al. (2013) have also imaged a subglacial outlet channel incised into the underlying sediment, which probably drains melt water flow or episodic floodwater from subglacial lakes. Similarly,

there might be a subglacial outlet channel crossing the sediment bed of the KIS trunk estuary towards the narrow trough shown in Figure 7. This speculation implies the possibility of sheet (distributed) flow and channelized flow coexisting beneath the estuary.

Recent studies (Alley et al., 2016; Le Brocq et al., 2013) have proposed a plausible physical mechanism to explain a line-shaped trough pattern on the ice shelf surface: they have suggested that when the subglacial meltwater flows into sub-ice-shelf cavities, producing meltwater plumes, heat from entrained ocean water in the plume can induce channelized melting in the underside of the ice shelf (Jenkins, 1991). A similar mechanism is proposed by Marsh et al. (2016) for the WIS subglacial channel outflow area. If this mechanism is operating at the KIS, it is probable that the strong basal melting by the outflow of its subglacial water system could create a similar channel there. In this conceptual model, the channelized area is likely to migrate upstream (and erode the grounded ice, expanding the sub-ice-shelf cavity in the estuary region) rather than advancing downstream, because the ice sheet flow toward the ice shelf at the lower KIS is very slow (~7 m/yr). The steep ice base near the grounding line of the KIS trunk estuary (as observed in BEDMAP2) may support the strong basal melting by meltwater plume (as described in Jenkins (2011)). Consequently, we speculate that the feature shown in Figure 7 is associated with a subglacial meltwater channel linked to the sub-ice-shelf cavity. Since other similar features are not observed around the grounding line of the KIS, we believe that the present-day KIS trunk has a single subglacial channel that reaches the grounding line, as previous studies have predicted (Carter and Fricker, 2012; Goeller et al., 2015).

Previous hydrologic models and observations in the upstream KIS show the possible diversion of basal water to the neighboring WIS (Anandakrishnan and Alley, 1997; Carter and Fricker, 2012), supporting the water-piracy hypothesis for the stagnation of the KIS (Alley et al., 1994). Because the volume changes of the upstream lakes K1 and K34 are significantly larger than those of the SGLs in the trunk of the KIS, the water-piracy hypothesis might be indirectly supported here. However, any connection between the upstream lakes in the KIS and the SGLs in the WIS is not clear from our observations. Comparison of our results to those of Siegfried et al. (2016) only confirm that the subglacial water system in the WIS is more active than the KIS trunk subglacial lake system, since volume changes in the WIS system are 3–10 times as large as in the KIS trunk system. Marsh et al. (2016) reported extremely large melt rates (22.2 ± 0.2 m/yr) at the site of inferred subglacial water discharge at the Whillans/Mercer Ice Stream grounding line. Strong basal melting is also inferred from the elevation loss rate (up to ~1.5 m/yr) in the narrow channel near the grounding line of the KIS, but the basal melt rate cannot be exactly projected from the observed elevation lowering due to the possibility of bridging stresses across the narrow channel feature.

**5 Conclusion**

We infer the presence of previously undiscovered subglacial lakes in the KIS trunk on the basis of localized elevation changes at sites of low hydraulic potential. Moreover, the subglacial lakes appear to be relatively tightly connected by channelized flow, following paths predicted by the hydraulic potential field, and respond in sequence to the apparent

input of water from a known lake system upstream of the KIS trunk. At the inferred outlet of the channelized flow system in the sub-ice-shelf cavity at the grounding line, a rapidly-eroding channel has been formed. We conclude that this is due to enhanced thinning of the ice sheet, and rapid basal melting at the meltwater outlet by entrainment of ocean water in a plume initiated by the freshwater outflow.

5         One hypothesis explaining the stagnation of the KIS is a conversion from sheet basal water flow to channelized flow, which leads to dewatering of the subglacial deforming till, thus immobilizing it (Retzlaff and Bentley, 1993). The subglacial water flow investigated in this study, and the inference of channelized flow beneath the KIS, is consistent with this conceptual model. However, the fact that the activity of the SGLs in the KIS trunk is much lower in total volume and flux than the discharge of the upstream lakes on the KIS may support a water-piracy hypothesis (Alley et al., 1994;

Anandakrishnan and Alley, 1997). Our results cannot definitively determine which phenomenon has dominantly effected to the stagnation of KIS. Both mechanisms could be active underneath the KIS. Further studies on the hydrological connections between the KIS and adjacent ice streams will be necessary to understand the stagnation and future evolution of the KIS.

*Acknowledgements.* This work was supported by a Korea Polar Research Institute (KOPRI) project (Grant No. PM16020) funded by the Ministry of Oceans and Fisheries, Korea; the Korean National Research Foundation grant NRF2013R1A1A2008368 Brain Korea 21 Plus Project in 2016; and the U.S. National Science Foundation grant PLR-1565576. The CryoSat-2, ICESat, and Landsat data used in this study are available from the European Space Agency, the National Snow and Ice Data Center, and the U.S. Geological Survey, respectively. We acknowledge two anonymous

reviewers for useful comments which substantially improved this paper.

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

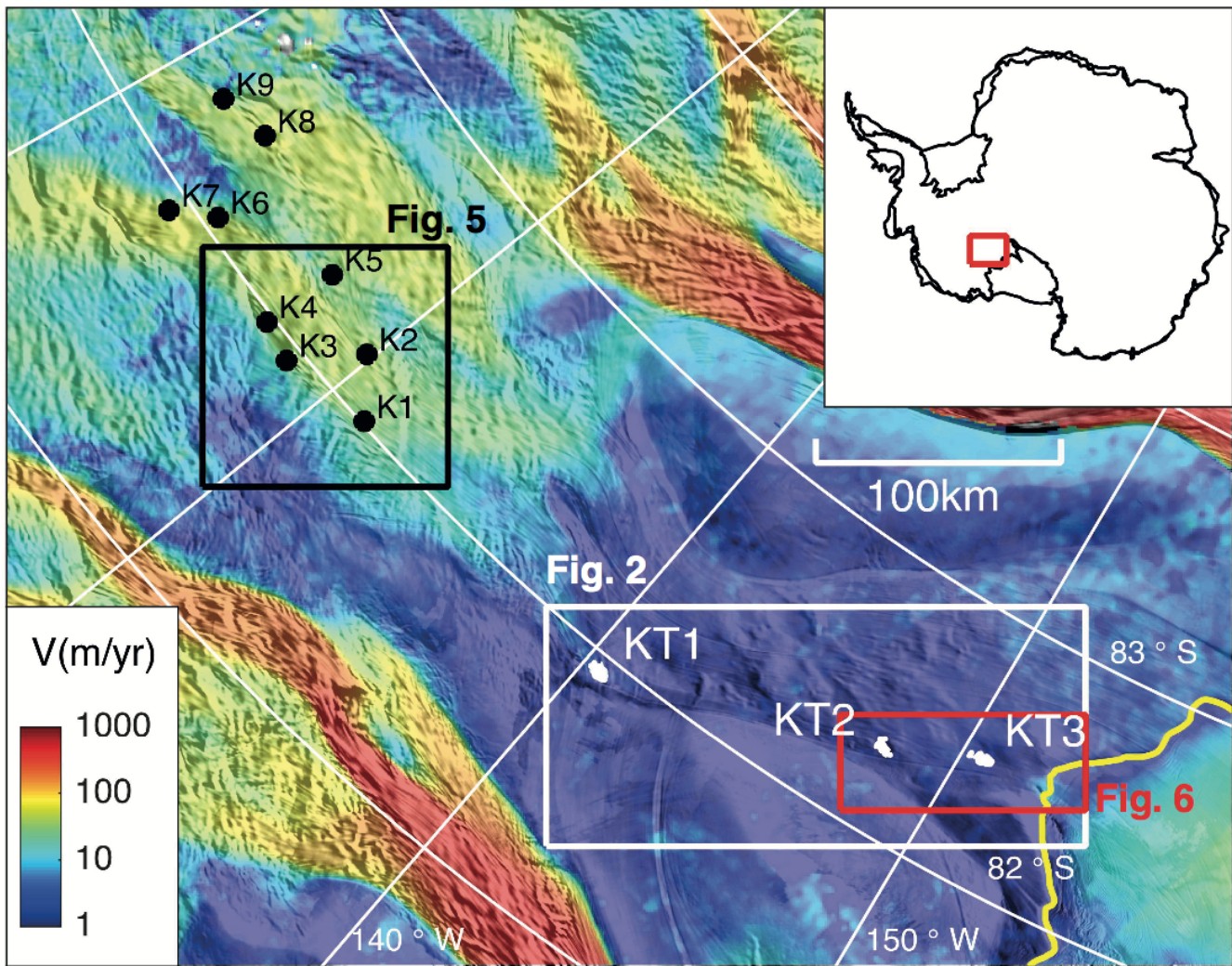

**Figure 1: Locations of subglacial lakes (Smith et al., 2009) in the KIS glacial catchments including the newly identified lakes KT2 and KT3 in the trunk of the KIS. The color shading shows the InSAR-based ice velocity from MEaSUREs (Rignot et al., 2011b) on the MOA image (Haran et al., 2014). The white, black, and red rectangles indicate the areas shown in Fig. 2, Fig. 5, and Fig. 6 respectively. The yellow line is the grounding line from Bindschadler et al. (2011). The lake outlines (white solid lines around the KT lakes) are estimated in this study.**

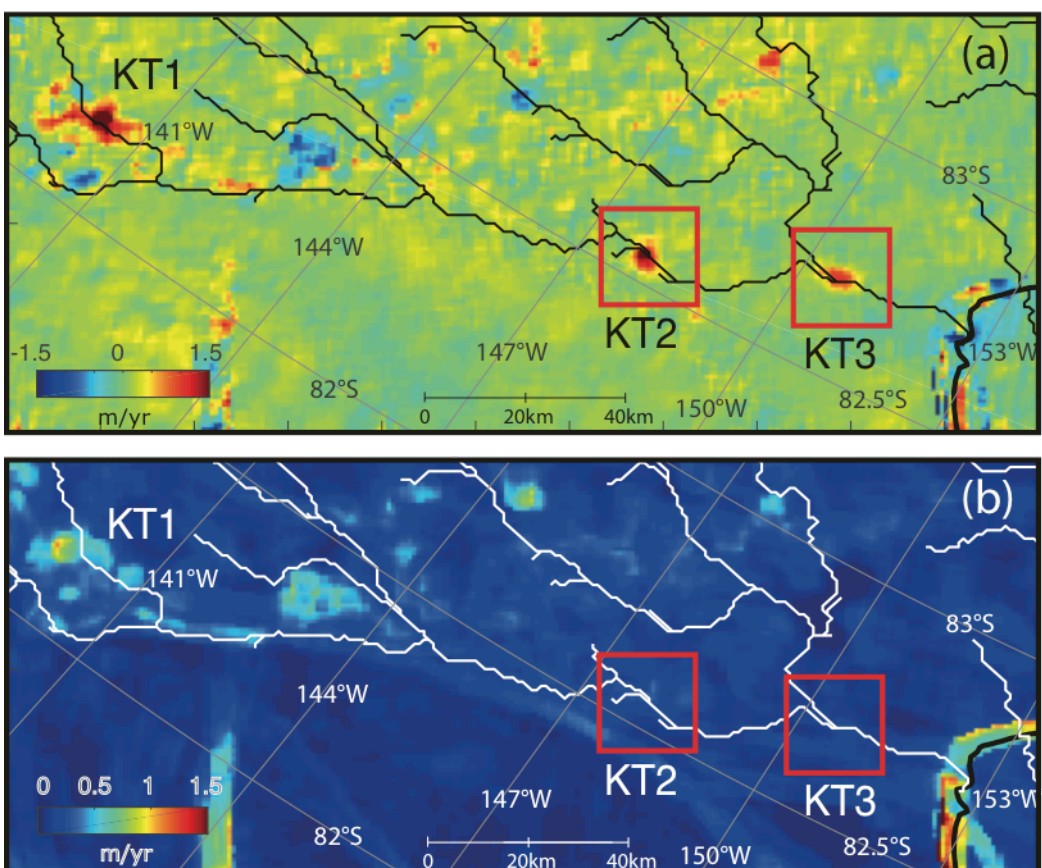

**Figure 2: (a) Elevation change rates and (b) their uncertainties (95% confidence intervals) around the trunk of KIS for two years**
10   **(February 2012 – January 2014). The polylines represent the subglacial streamlines extracted from the hydraulic potential. The red boxes indicate the locations of lakes KT2 and KT3.**

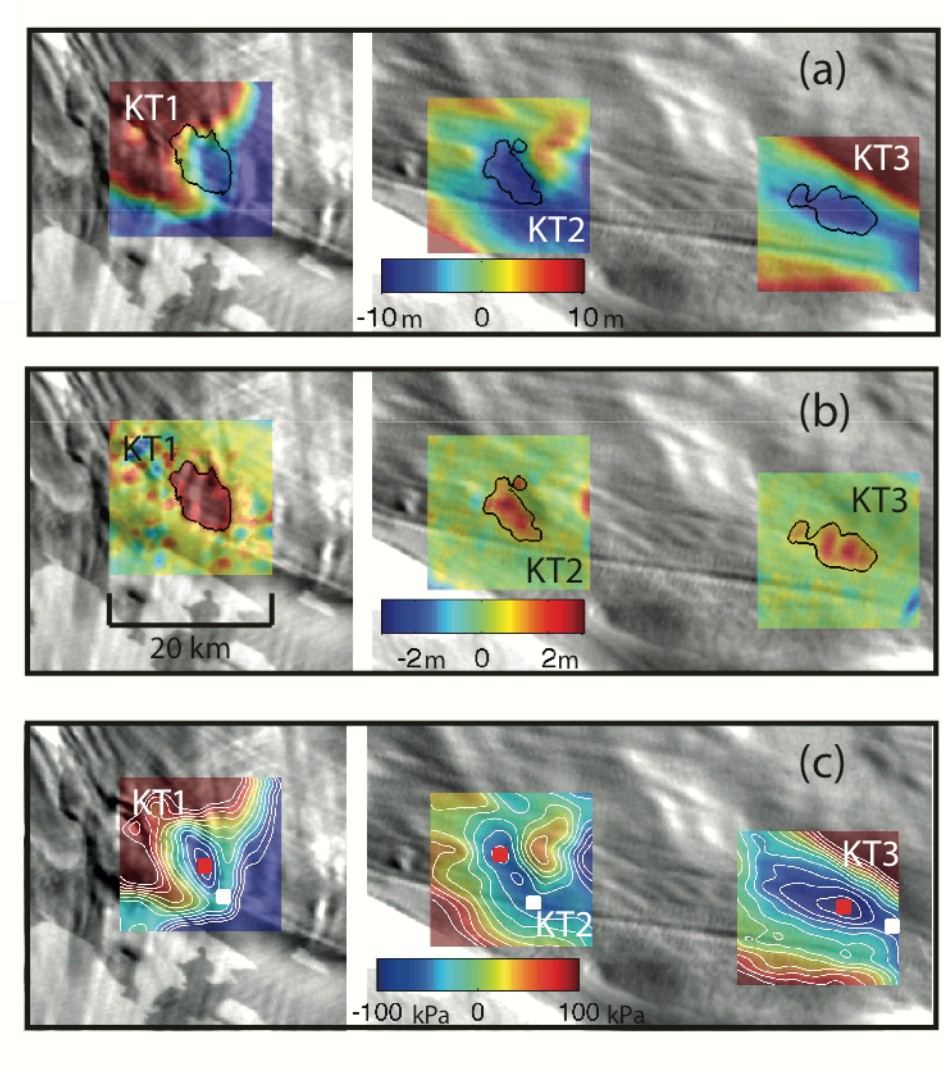

**Figure 3: Ice surface elevations and subglacial hydraulic potentials around the three subglacial lakes KT1, KT2, and KT3, overlaid on the MOA image. (a) Reference DEMs (color shading) derived by the kriging method using the CryoSat-2 elevation measurements. (b) Elevation anomalies deviated from the reference DEMs in the time windows as described in the text. (c) Relative hydraulic potential anomalies calculated using the reference DEMs and ice thickness from Bedmap2. The contour lines are plotted at 20kPa intervals. Red squares indicate the centers of the hydropotential lows, and the white squares indicate the predicted outlets. Note that the mean values of elevation and hydraulic potential in each rectangular area are subtracted in order to reveal the details.**

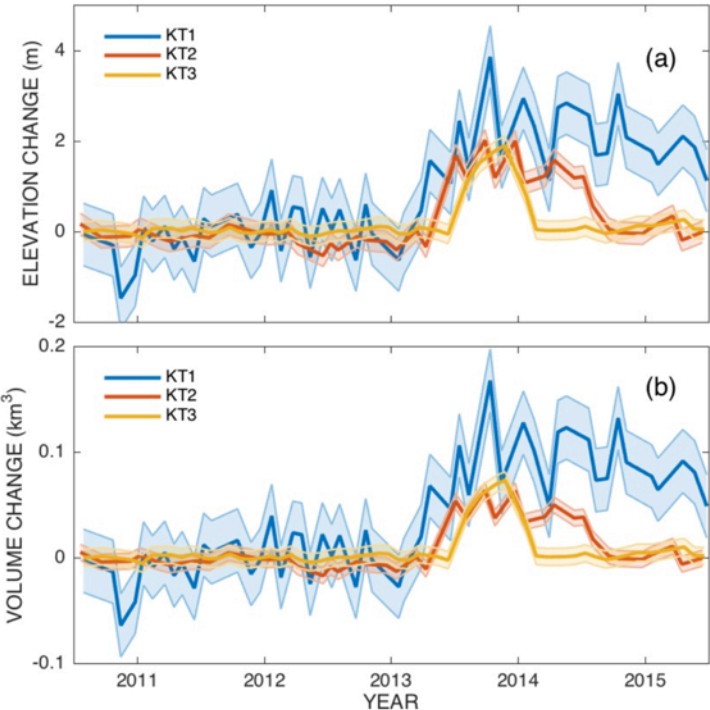

**Figure 4: Temporal (a) elevation and (b) volume changes of subglacial lakes in the trunk of the KIS. The error ranges are displayed by transparent coloring.**

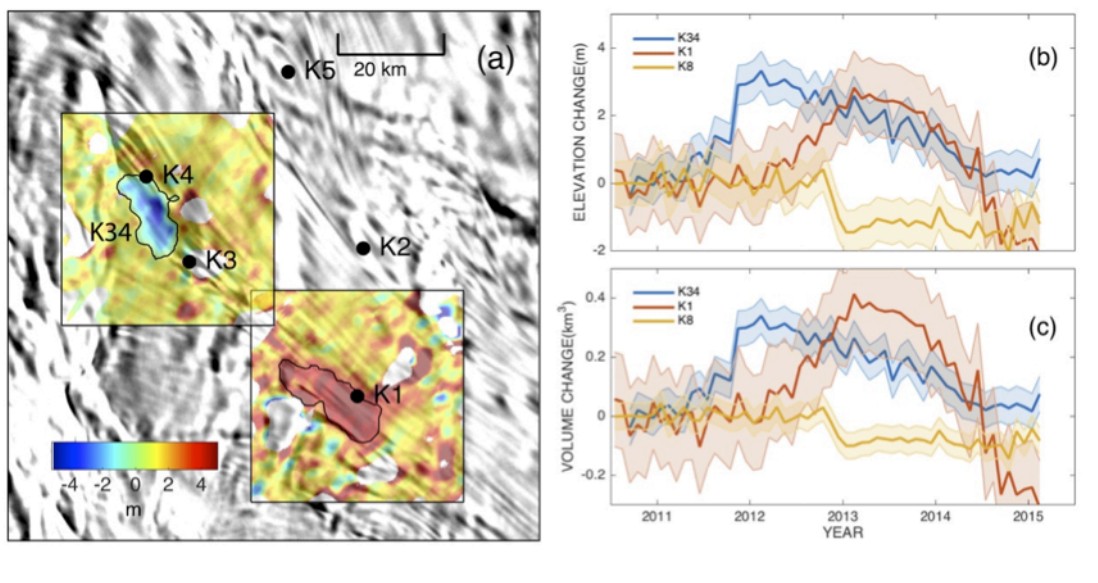

 **Figure 5: Spatial and temporal changes of the subglacial lakes in the upper region of the KIS. (a) DEM difference (color shading) around the subglacial lakes K1 (2013 – 2011) and K34 (2014 – 2012). The difference in the area with large kriging uncertainties (>90%) is masked. The polygons indicate the lake boundaries. The black dots are the locations of the lakes already listed in Wright and Siegert [2012]. The right-hand panels show the temporal (b) elevation and (c) volume changes of the lakes. See Fig. 1 for the location of lake K8.**

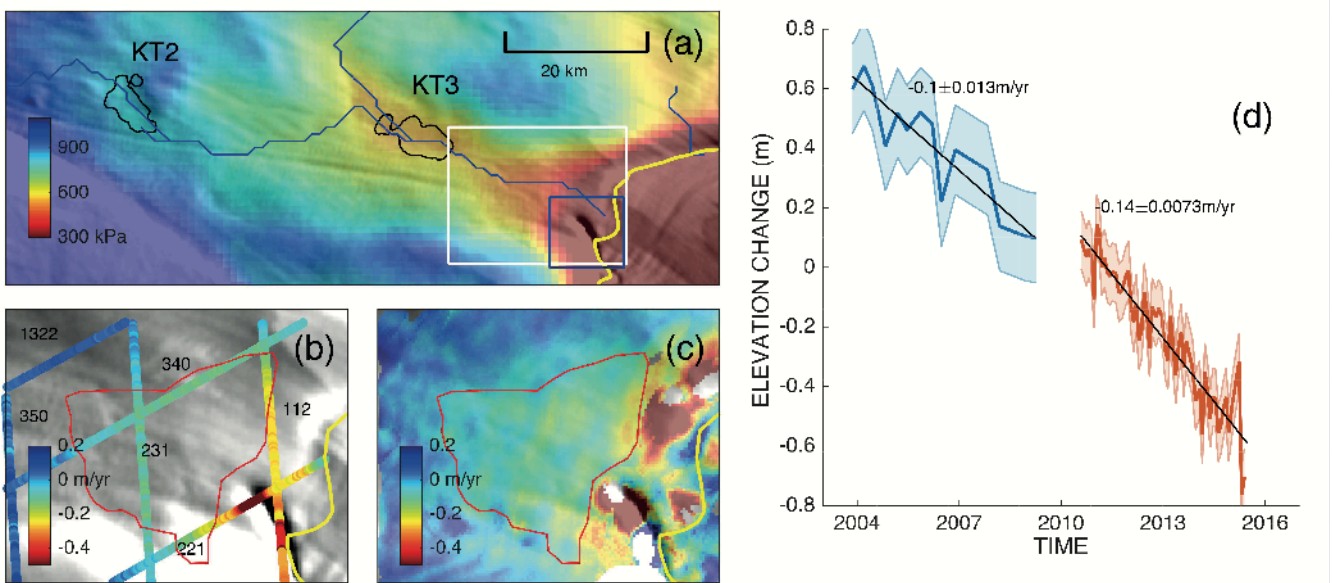

**Figure 6: Elevation changes near the KIS grounding line. (a) Hydraulic potential (color shading) and flow lines (blue lines) from**
**KT2 to the grounding line (yellow line). The white rectangle indicates the area displayed in (b) and (c). The blue rectangle**
**indicates the area shown in Figure 7. (b) Elevation change rate estimated from ICESat repeat-track analysis. (c) Elevation change**
**rate estimated from CryoSat-2 DEM differencing (2014–2011). (d) Temporal elevation changes of ICESat and CryoSat-2**
**measurements in the 'estuary' area, denoted by the red polygon in (b) and (c). The bias of Cryosat-2 elevations was removed,**
**including the instrument bias (0.67 m) and the effect of radar penetration into the snow pack (1.03m) which is estimated along an**
**adjacent ICESat track.**

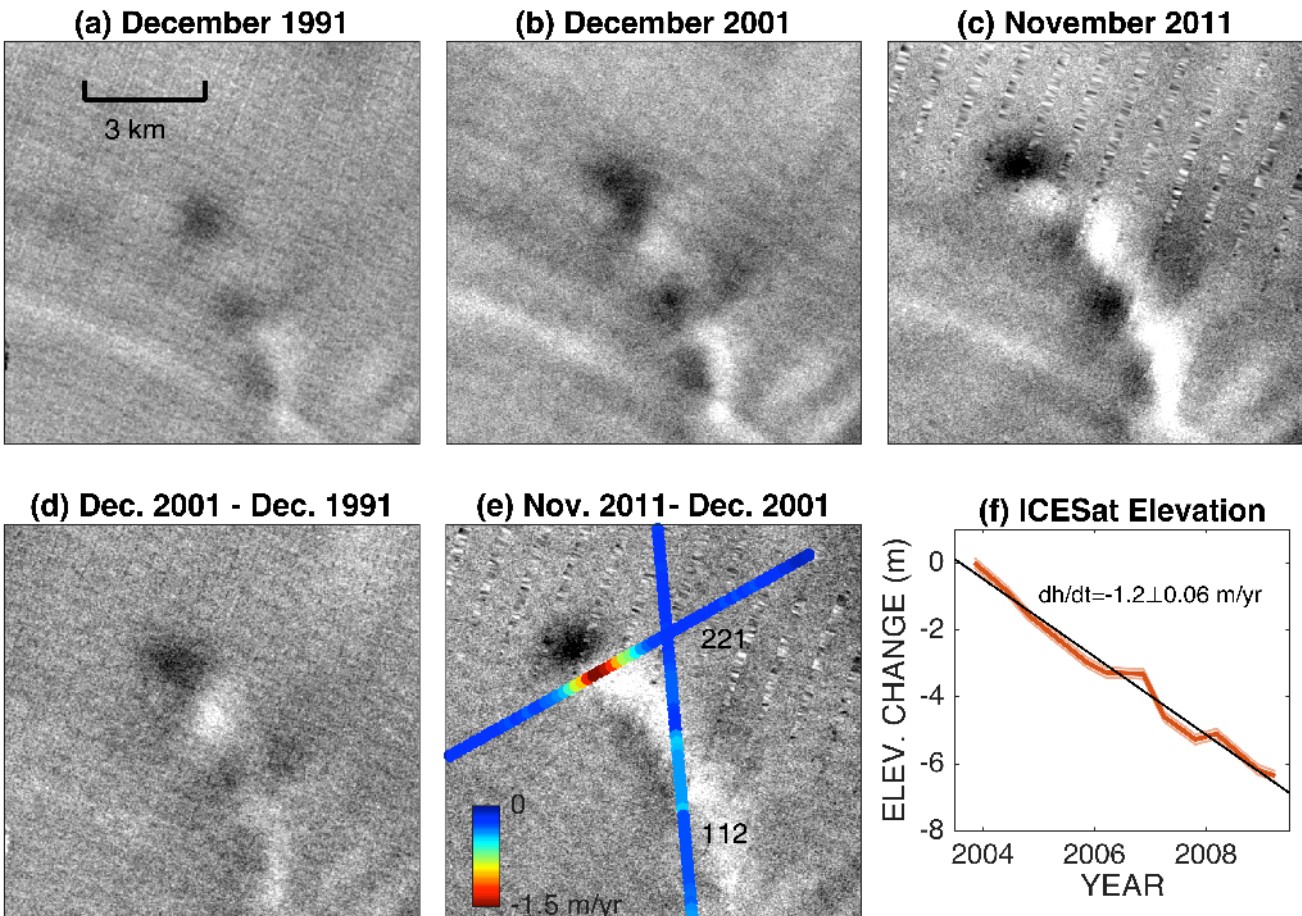

**Figure 7: Landsat images ((a) – (c)) and their difference images ((d) – (e)) over the region indicated by the blue box in Figure 6. The stripes in the image for November 2011 due to the failure of the scan line corrector (SLC) of Landsat-7 ETM+ are removed by a gap-filling method using additional images taken around the same time. The elevation change rates from the ICESat measurements are overlaid on the difference image (e). Panel (f) shows the temporal elevation change in the topographic sink, measured along the 221 ICESat track.**