# Peer review of "Active subglacial lakes and channelized water flow beneath the Kamb Ice Stream"

_The Cryosphere, 2016_

## Referee Comment (RC1) · Anonymous Referee #1 · 13 Jun 2016

This is interesting work, which extends our knowledge of the processes controlling the dynamical behaviour of the Kamb Ice Stream. I believe this work has the potential to make a substantial contribution to our field. However, there are several fairly major issues that need to be addressed ahead of publication; these are outlined below.

General points:

1. I appreciate that for 4 out of the 5 authors, English is not a first language, and although the language is fairly good (much better than I could do in Korean), there are still many instances in the manuscript that need to be edited for English grammar/language – I would recommend the native English speaking author to edit the manuscript accordingly. 2. I found the structure of the paper to be a little confused – methods where in

the results, discussion in the results, no conclusion section. This needs to be rectified with careful attention as it makes the manuscript difficult to follow. 3. I am confused as to why you present so many methods for deriving elevation change from CryoSat-2 data – I think there are three – one of which was introduced in the results section (see note above). Is this necessary? And if you believe so, please be clear why it is that you are doing this and which results are linked to which method. 4. You introduce ICESat results in the results section but there is no mention of this in the methods section. You need to include a description of this data and the processing methodology in you methods section. 5. Throughout your text you report numbers of surface lowering from altimetry but you do not report associated errors – this makes it difficult to discern whether or not the changes in rates that you report are significant or not. I suspect in some instances they might not be. This must be rectified. 6. Also regarding errors, you report in you methods that you output and error statistic for the CryoSat elevation change signals from the regression error but later, in figures, use uncertainty measures from standard deviations of measurements and also from the signals from (assumed) constantly changing regions. I think it would be better/clearer/less confusing to stick to just one measure of uncertainty for any given parameter.

Specific points (given by Page number and Line Number – e.g. P1 L1):

P1 L112 – change "inferred from CryoSat-2 altimetry, indicating that" to " inferred from surface height changes using CryoSat-2 altimetry and indicate that. . ."

P1 L13 – change "The orientation of the drainage network. . ." to 'The structure of the subglacial drainage network. . ." P1 Introduction – general comment: I think that in the introduction you need to explain more about the previous work done into understanding the stagnation of the KIS. There is more than one theory for why this ice stream stagnated, and so I think that you need to acknowledge this.

P1 L20 – this is a bit of an empty statement – explain why or to what extent the stagnation is significant for the mass balance of the WAIS – i.e. there is a mass gain.

P2 L5 – you outline the limitations of the ICESat data, but what about the limitations of interferometry (which you mention in Line 3) for detecting SGLs ?

P2 L5 is this lake in the Northern Corner highlighted in any figure? If not, I would do so and include a reference here.

P2 L10 – it would be helpful if you clearly stated the aims and objectives of your study here.

P2 L 29 – why not just use a simple threshold at this stage?

P3 L3 – Why not just use a smaller grid cell size? It is likely that there would be sufficient data within the grid cells at $\sim$ 2 km to provide a robust solution.

P3 L5 – regression of what? Do you mean the quadratic fit?

P3 L6 – is your selection of lake areas subjective or quantifiable? You need to explicitly state what criteria you use to determine if an area is lake or not.

P3 L9 -10 – these are results not methods

P3 L10 – I would be interested to know if there are any potential lakes where you don't see any.

P3 L11 – I don't understand why you have also use the plane fit method if this provides more detailed information on the boundaries. Also, I don't really understand how this can provide more detailed information on the boundaries, because the same data should be included in both methods.

P3 L12 state the resolution of these DEMs. Also what filtering criteria, if any, do you use to remove unreliable height estimates.

P3 L18 – why do you think this is? Provide a physical explanation.

P3 L18-19 – lake areas should be in results.

P3 L19-20 – more detail needed for the methodology of deriving the hydraulic potential

– at least a reference to the paper(s) from which you derived the method.

P3 L25-29 – this is methods.

P3 L27 – state the magnitude and standard deviation of the trend removed? How much spatial variability was there in this trend?

P3 L 32- include uncertainty – this needs to be done for all your results (numbers) reported in the text.

P4 L 4 "descend completely yet" – replace with " returned to the previous level by the end of the study period."

P4 L7 – L17 – this is all discussion

P4 L 20 – what are the values of head differences when the lakes are fully filled? Include numbers.

P4 L28 – you need to include your methods for using the LRM data in your methods section.

P4 L 35 replace "present" with 2015.

P5 L 9 significant effect on what?

P5 L 9 – what exactly do you mean by the estuary? What characteristics does it have? In Figure 6 you show the area in a box – on what basis have you defined this area?

P5 L11 – where has this ICESat data come from? Again you need to include this in your methods.

P5 L 12 – How have you accounted for mission offsets in the elevations measured by the two missions? If you haven't, I don't think that you can meaningfully present the two as a 'continuous' time series on the same plot. Suggest you either show on separate plots, or show one plot of elevation changes (dhdt).

P6 L1 – explain why the feature is too concave for CryoSat-2 to detect.

P6 L8 – Change "If this method is applicable to the KIS, the strong. . ." to " If this method is applicable to the KIS, it is probable that the strong. . .."

P 7 – what about the conclusion section?

Figure 1 – I presume the yellow line is the Grounding Line? Include in figure caption.

Figure 2 – There are two regions of surface lowering and 1 region of surface increase in the north of this figure? Why aren't these identified as lakes?

Figure 3 – why not show continuous fields of the ice surface elevations and elevation anomalies?

Figure 4: why do you show the standard deviations here and not the uncertainty on the elevation change measurements that you derived in the processing?

---

## Referee Comment (RC2) · Anonymous Referee #2 · 3 Jul 2016

This manuscript documents the discovery of two new subglacial lakes beneath the currently stagnant portion of Kamb Ice Stream, West Antarctica, from CryoSat-2 satellite radar altimetry and ICESat satellite laser altimetry. This is a novel observation and extends our knowledge of subglacial hydrology and ice dynamics in this region. Additionally, it provides context for understanding centennial-scale ice-stream stagnation and reactivation cycles, which are currently an important control on ice-sheet mass balance in this sector of West Antarctica. Finally, this work provides an example of using decades of observations of altimetry data – data which should be used more often – to understand subglacial hydrology and dynamics. This work has the potential to substantially add to our knowledge of subglacial hydrology and presents

new techniques for analyzing long time series of satellite altimetry data. However, as currently written, there are organizational and scientific issues that must be addressed before publication

Please also note the supplement to this comment:
http://www.the-cryosphere-discuss.net/tc-2016-96/tc-2016-96-RC2-supplement.pdf

**Supplement:**

**Summary**

This manuscript documents the discovery of two new subglacial lakes beneath the currently stagnant portion of Kamb Ice Stream, West Antarctica, from CryoSat-2 satellite radar altimetry and ICESat satellite laser altimetry. This is a novel observation and extends our knowledge of subglacial hydrology and ice dynamics in this region. Additionally, it provides context for understanding centennial-scale ice-stream stagnation and reactivation cycles, which are currently an important control on ice-sheet mass balance in this sector of West Antarctica. Finally, this work provides an example of using decades of observations of altimetry data – data which should be used more often – to understand subglacial hydrology and dynamics. This work has the potential to substantially add to our knowledge of subglacial hydrology and presents new techniques for analyzing long time series of satellite altimetry data. However, as currently written, there are organizational and scientific issues that must be addressed before publication.

**General Points:**

1) **Organization/structure** – This paper at times lacks a clear organizational structure with introductions, methods, results, interpretation/discussion, and conclusions. I think that following this traditional structure could be useful in this case, as many different methods are being used and it is often difficult for the reader to determine the reason for selection of the method or analysis technique applied.

2) **Methods and uncertainty** – ICESat laser altimetry, surface elevation, and bed elevation data are used throughout the paper with clear explanation or citation. A thorough methodology and uncertainty analysis should be added to the manuscript. Regression error and standard deviation of residuals over non-changing portions are both used as measures of uncertainty. What are the reasons for including both uncertainty methods?

3) **Labeling** – Labeling in figures is often inconsistent. Sometimes panels are labeled; other times they are not. I'd prefer to err on the side of caution and label. Similarly, features in the figures are also often unlabeled or annotated. For example, grounding lines are not labeled or cited, but are plotted in the figures. Lakes outlines are also not consistently labeled, so it's unclear to me sometimes whether they are from Smith et al. (2009) or Wright and Siegert (2012) or this study.

**Scientific Points:**

1) **ICESat** – ICESat laser altimetry data are extensively used but are not discussed. This is especially crucial as these data are quantitatively compared to CryoSat-2 data, and the amplitudes of the elevation change derived are similar to the uncertainty of these methods. More discussion of the ICESat data and comparison between the ICESat and CryoSat-2 data would be useful to ascertain the significance of the signals analyzed.

2) **Hydropotential** – It is not clear what reference datum is used. It also seems as if two different datums are used as the hydropotential values differ significantly in the two figures shown. Finally, I assume that glaciostatic hydropotential (subglacial water pressure equal to overburden pressure) is calculated, but this is never stated.

3) **Subglacial lake detection algorithm** – I would expect a more precise outline of the steps used, similar to the algorithm presented in Smith et al. (2009). For example, it's unclear to me what "visually inspect" means.

4) **CryoSat-2 data** – Several methods are presented for analysis of the elevation data (quadrature curvature surface fitting, differences from reference DEMs, etc.). It's a little difficult to determine the exact sequence of methods being used and why. A more stepwise description and possibly flowchart figure would add clarity.

**Style/Lanuage/Grammar comments**

1) Hyphenation use is inconsistent – For example, "repeat track method" vs. "topography-free elevation" and "sub-glacial" vs. "subglacial". I tend towards hyphenation, i.e. repeat-track and topography-free, but use should be consistent.

2) Capitalization use is inconsistent – For example, Antarctic ice sheet vs. Siple Coast Ice Streams. I would tend to lean towards Siple Coast ice streams and Antarctic ice sheet, but capitalization should be consistent regardless of convention.

**Specific Comments**

P1 L11: "We have identified two previously unknown active subglacial lakes…"

P1 L12: Rapid fill-drain events do not necessarily indicate lake connectivity via a drainage network though they do indicate a subglacial drainage network exists.

P1 L13: "lakes area" seems redundant. Perhaps just "lakes."

P1 L14–16: This sentence should be rewritten to highlight the evidence that links subglacial lake drainages to the acceleration of thinning.

P1 L15: "subglacial lakes".

P1 L16–17: It seems unlikely to me that this conclusion can be justified from data presented here, which is too short a time series to suggest regions for the shutdown. It could be consistent with such a shutdown mechanism. This sentence should be reworded accordingly.

P1 L16: Figure 2a does not seem to indicate rapid thinning. If anything, thickening seems the dominant signal. Perhaps the background elevation-change rate has been removed? Hopefully stated later.

P1 L17: "sub-glacial" to "subglacial".

P1 L19: Change "rapid ice flow" to "streaming ice flow".

P1 L20–21: This sentence reads awkwardly. This paragraph could perhaps be restructured to facilitate reading. I would suggest the following general order:

1) Introduce ice-stream stagnation/reactivation cycles and their effects on ice-sheet mass balance.
2) Note that KIS stagnated ~160 years ago.
3) Note that changes in basal hydrology (rather via water piracy or change in dominant subglacial hydrology system structure) are likely a cause of this stagnation.

P1 L21: Change "indicated" to "posited", "hypothesized", "suggested", or similar to indicate hypothetical nature of this conclusion (even though it is likely correct).

P1 L21: Change "Siple-coast" to "Siple Coast".

P1 L22: Change "Ice Stream" to "ice streams".

P1 L22: Delete "of these cycles".

P1 L24: Change "long term" to "long-term".

P1 L24–25: Presumably it is associated with changes in basal-melt rate and upstream subglacial water supply, but how are these related? Are the authors suggesting a Tulaczyk et al. (2000) thermodynamic till mechanism or purely water piracy?

P1 L25: "Therefore, it has been suggested…"

P1 L28: Change "predict its dynamics" to "understand the ice dynamics" or similar.

P1 L30: Change "while" to "although".

P2 L1: Delete "contributing the basal hydrology".

P2 L3–4: "the hydrological connections between adjacent lakes".

P2 L4: "by the sparse coverage of the ground tracks".

P2 L5–7: Is this a supposition of the authors or are there other studies that posit this?

P2 L11–16: This paragraph mixes methods and introductory materials.

P2 L11–12: Inconsistent capitalization: Antarctic Ice Sheet here vs. Antarctic ice sheet previously.

P2 L13: "two previously unknown subglacial lakes"

P2 L14: What sort of activity? Downstream lakes fill as the upstream lakes drain?

P2 L15: Change "evidences" to "evidence".

P2 L13–15: This sentence is awfully general. Perhaps change to something more specific to the results presented in this manuscript.

P2 L17: Seems like there should be a more extensive methods section.

P2 L21: "geophysical" should probably be "geographic".

P2 L23: Which version of the L2 product? If I remember correctly, there are multiple processing baselines.

P2 L24: What does "interior of ice" mean? Ice-sheet interior? Why the range of uncertainty?

P2 L24–26: I am confused by this sentence. There seem to be multiple numbers for uncertainty and their dependence on physical values varies. A more complete uncertainty approach is probably needed, or at least a more complete discussion of the approach adopted here.

P2 L27–P3 L10: I think a step-by-step description and/or processing flow chart is needed here. As is currently written, the description of the processing steps is somewhat unclear.

P2 L28: What do height error flags signify?

P2 L30: Does this method follow Helm et al. (2014) or is it distinct in some way from that processing?

P3 L3–5: What does visually inspect actually mean? To be robust, I think quantitative metrics are required to determine reliability of subglacial lake detection. A protocol like that described by Smith et al. (2009) seems to be needed here.

P3 L5: What are the uncertainties estimated from the regression? It is hard to ascertain what the numbers actually are from Figure 2b.

P3 L6: What is sufficiently lower? A more rigorously quantified uncertainty analysis is needed.

P3 L12: In various time windows? Different windows seem to represent different lengths of time? I would have thought an overlapping window scheme of a constant time length would be used? If these irregular windows can be easily justified, I'd like to know why.

P3 L19: I assume this is glaciostatic hydropotential following Shreve (1972), but it would be nice to have this verified at least once.

P3 L19–20: Why not say directly that the "lakes are located in hydropotential lows."

P3 L24–25: This sentence does not seem necessary.

P3 L27–28: "The background temporal elevation changes outside the lake boundaries are removed to examine only the elevation changes associated with the SGLs activity." How was this done precisely?

P4 L1: Are the elevation change numbers sufficiently robust to actually derive this balance flow rate? What is the associated uncertainty?

P4 L1–4: This reverse sequential drainage seems odds compared to the sequential filling. If KT1 is supplying the other two lakes, why does it not continue filling if the water is no longer flowing to the other lakes? Does it go someplace else? Or has the inflow rate just slowed? In Figure 4, the fluctuations seem large relative to the uncertainty. Is the uncertainty here truly representative of that in these data? Can these high-amplitude fluctuations be interpreted?

P4 L5–15: Much of this seems speculative and just descriptive of the results of other studies. Are the flux rates from these lakes sufficient to maintain a connected network of Röthlisberger channels as the authors seem to suggest here? Alternate theories also seem possible. The simplest explanation may be that these lakes, separated by ~100 km, simply are not hydraulically connected and fill and drain separately. I am hesitant to put much faith in hydropotential maps on this level and much of the inflow from KT3 appears to come from a separate upstream hydraulic catchment anyway. If the authors are suggesting that a channelized hydrology system can be maintained against creep closure via this water flux, I'd like to see some calculations and type of channel suggested (R- or N-channel, canal network, etc.). I note that the one real-time observation of transient water flow under an ice stream (i.e., not a subglacial lake filling or draining, but subglacial water in motion; see Winberry et al. (2009)) indicates that the channels are probably not maintained for more than a few weeks, not the many months suggested here.

P4 L19: It is hard to verify the hydraulic head difference cited here from the figures. Looking glibly at Figure 4, the hydraulic head changes seem like they should be somewhat less than the numbers cited here. However, it's difficult to tell if the authors are including flow focusing or some other effect in their calculation. More details are needed.

P4 L23: "role of the hydraulic barrier" or "role of hydraulic barriers".

P4 L25–28: This sentence could be split into two or a comma should be added to separate clauses: "…KIS area, but the…".

P4 L28: How are the LRM products used? This should be added to the methodology, or at least there should be a citation if following an established method. LRM mode differs significantly from SARin, and thus differences in uncertainty, reliability, resolution, etc., would be expected; clarification of these differences is needed.

P4 L32 – P5 L1: This is more like the patter of draining I would expect. I wonder if an extended discussion of the difference between the upstream and downstream lakes would be useful.

P5 L8–9: This sentence seems like a conclusion before the data/interpretation/discussion are presented.

P5 L12: Methods on how the ICESat and CryoSat-2 time series are combined are necessary. To do this properly, error and biases should be clearly presented.

P5 L15–17: Can you really assess when precisely KT1 stopped draining with the elevation amplitude presented in Smith et al. (2009). I would think the ICESat data would allow more precise determination of subglacial lake drainage timing. A drainage lasting this long seems unlikely compared to other subglacial lake drainages documented (c.f., Siegfried et al., 2016).

P5 L19–22: How does this estuary compare to the one documented on Whillans Ice Stream (see Horgan et al. (2013a,b) and Christianson et al. (2013)). A discussion of similarities and differences between the estuaries would be important, as progradational till deltas were directly observed in that estuary and it seems as the authors are suggesting a similar depositional structure here, with sheet (distributed) flow and channelized flow coexisting.

P5 L23–24: Enhanced lubrication isn't a necessary condition for this. Tensile forces must inherently exist at the junction of an ice stream and ice shelf (see Weertman (1974) and Schoof (2007) among others). Although additional basal lubrication could result in increased longitudinal stress. The timing of the lake drainage does seem nicely correlated with the increase thinning, but it could have but both events could have been triggered as a result of regional grounding line retreat, and thus the lake drainage could have been an effect and not a cause. Some discussion of the nuances and limitations of the data would be helpful, as well as connecting more directly to known background thinning rates and ice bottom/bedrock geometry.

P5 L30: These feature looks distinct from the channel described in Marsh et al. (2016), which does not have undulating topographic features in the along flow direction. Some discussion of why this might be would be useful.

P5 L33: I am suspicious of the 30 m/yr retreat rate. Grounding-line location was not well known along the Siple Coast in the late 1980s. Is there no newer result?

P6 L6: "to the oceans".

P6: L4–8: There should probably be a citation to the buoyant meltwater plume circulation that drives this circulation in the channel – I'd suggest Jenkins (1991) and Jenkins (2011). Ruling out other possible channel creation mechanisms is probably needed too, i.e., highly variable bed topography, suture zones, etc.

P6 L12–13: Is the "observed" channel in the same location as the modeled ones?

P6 L15: Here and throughout the manuscript I wonder if the distributed rather than sheet flow might be what the authors are describing. Sheet flow implies a thin water film a few millimeters thick. Distributed flow would allow more generality.

P6 L19–21: Much of this seems like conjecture. Other drainage systems besides a strictly channelized system could lead to relatively rapid connectivity between lakes. Small outburst floods with transient channelization (see Winberry et al. (2016)) seem more likely to me than a long-lived R-channel connecting lakes. The flow of water along paths down the hydropotential gradient does not necessarily imply channelized flow either.

P6 L30: Perhaps "Comparison of our results to Siegfried et al. (2016)" to avoid confusing the data presented here with those derived from studies of Whillans Ice Stream subglacial hydrology.

P7 L5: "definitely" should perhaps be "definitively"?

P7 L5–8: Perhaps reword as: "At present, our results cannot definitively determine whether KIS stagnation occurred via basal water channelization, water piracy, or some combination of these. Further studies of basal water flow in the KIS trunk would be necessary to make this determination." I don't see the need for channelization or water piracy to be mutually exclusive.

Figure 1: Smith et al. (2009) or other sources should be cited for location of previously known subglacial lakes. Grounding line (yellow) and appropriate citations should be given in the caption. What is the date of this grounding line? Citations for MEaSUREs velocity data and MOA imagery are similarly needed (I cannot tell which version of each was used). Some reference should be made to lake outlines shown for KT1, KT2, and KT3 (even if "as discussed in the text."). Some demarcation should be shown for Figure 5–7 (or shown sequentially if not marked here). This is especially crucial for Figure 5, as there are not good references to scale for that figure.

Figure 3: What datum is used in hydropotential calculations. Although the gradients are the amplitude I'd expect, the value of hydropotential itself seems unlikely to be slow. Theoretically hydropotential should go to 0 at the grounding line. Although some values of hydropotential may drop below 0 on grounded ice, this seems to be widespread. I suspect this is a result of the datum being used.

Figure 6: Once again, hydropotential values seem off. They should go to near 0 near the grounding line. Even using a standard datum (WGS84 ellipsoid; EGM2008 geoid) would get relatively close. These seem too high. These values are also inconsistent from those shown in Figure 3. Grounding line (in yellow) should be noted in caption. How was the estuarine area shown in red determined? Why is there no uncertainty on the elevation change plot (d) when there are uncertainties on similar plots in other Figures? Label ICESat tracks shown in Figure b.

Figure 7: Though not particularly important, I am surprised gap filling did perform better for the November 2011 Landsat 7 scene. I assume the fit in the final panel is linear, but it would be good to note this. Uncertainty values in elevation-change panel would also be desirable. I suspect that labeling the panels (a–e) would be useful here too.

**References**

Christianson, K., B. R. Parizek, R. B. Alley, H. J. Horgan, R. W. Jacobel, S. Anandakrishnan, B. A. Keisling, B. D. Craig, and A. Muto (2013). Ice sheet grounding zone stabilization due to till compaction. *Geophysical Research Letters 40*, 5406–5411, doi: 10.1002/2013GL057447.

Horgan, H. J., K. Christianson, R. W. Jacobel, S. Anandakrishnan, and R. B. Alley (2013a). Sediment deposition at the modern grounding zone of Whillans Ice Stream, West Antarctica. Geophysical Research Letters 40, 3934–3939, doi: 10.1002/grl.50712.

Horgan, H. J., R. B. Alley, K. Christianson, R. W. Jacobel, S. Anandakrishnan, A. Muto, L. H. Beem, and M. R. Siegfried (2013b). Estuaries beneath ice sheets. *Geology 41*, 1159–1162, doi: 10.1130/G34654.1.

Jenkins, A. (1991). A One-Dimensional Model of Ice Shelf-Ocean Interactions. *Journal of Geophysical Research 96*(C11), 20,671–20,677.

Jenkins, A. (2011). Convection-Driven Melting near the Grounding Lines of Ice Shelves and Tidewater Glaciers. Journal of Physical Oceanography 41, 2,279–2,294, doi: 10.1175/JPO-D-11-03.1

Marsh, O. J., H. A. Fricker, M. R. Siegfried, K. Christianson, K. W. Nicholls, H. F. J. Corr, and G. Catania (2016). High basal melting forming a channel at the grounding line of Ross Ice Shelf, Antarctica. *Geophysical Research Letters 43*, 250–255, doi: 10.1002/2015GL066612.

Schoof, C. (2007). Ice sheet grounding line dynamics: Steady states, stability, and hysteresis. *Journal of Geophysical Research 112*, F03S28, doi: 10.1029/2006JF000664.

Shreve, R. L. (1972). Movement of water in glaciers. *Journal of Glaciology 11*(62), 205–215.

Siegfried, M. R., H. A. Fricker, S. P. Carter, and S. Tulaczyk (2016). Episodic ice velocity fluctuations triggered by a subglacial flood in West Antarctica, *Geophysical Research Letters*, doi: 10.1002/2016gl067758.

Smith, B. E., H. A. Fricker, I. R. Joughin, and S. Tulaczyk (2009). An inventory of active subglacial lakes in Antarctica detected by ICESat (2003-2008). *Journal of Glaciology 55*(192), 573–594.

Tulaczyk, S., W. B. Kamb, and H. F. Engelhardt (2000). Basal mechanics of Ice Stream B, West Antarctica: 2. Undrained plastic bed model. *Journal of Geophysical Research 105*(B1), 483–494.

Weertman, J. (1974). Stability of the junction of an ice sheet and ice shelf. *Journal of Glaciology 13*(67), 3–11.

Winberry, J. P., S. Anandakrishnan, and R. B. Alley (2009). Seismic observations of transient subglacial water-flow beneath MacAyeal Ice Stream, West Antarctica. *Geophysical Research Letters 36*, L11502, doi: 10.1029/2009GL037730.

---

## Author Comment (AC1) · 14 Aug 2016

**Response to Anonymous Referee #1**

**We really appreciate for taking the time to review our paper and the comments valuable to improve our paper greatly. We address the comments in the order of the review as bellows. We noted the pages and line numbers indicating the corrections in revised manuscript corresponding to the comments. In addition, we would like to apologize a mistake in the analysis of the thinning in the estuary region. Owing to the referees' comments, we carefully validated the elevation time series from Cryosat-2 especially in the estuary region and found that the time series indicating the acceleration of thinning since 2013 were contaminated by the background elevation change (e.g. change of surface mass balance), because we could identify such changes in elevation on nearby stationary ice (e.g. even on the Siple Dome). We removed the background elevation changes around the estuary (modified Figure 6) and the statements related to the acceleration of thinning by subglacial flood in 2013. However, we believe it does not affect the main suggestion and conclusion of this paper.**

**General points:**

1. I appreciate that for 4 out of the 5 authors, English is not a first language, and al- though the language is fairly good (much better than I could do in Korean), there are still many instances in the manuscript that need to be edited for English grammar/language – I would recommend the native English speaking author to edit the manuscript accordingly.
⇒ The revised manuscript would be improved. We hope that it can be read easily now.

2. I found the structure of the paper to be a little confused – methods where in the results, discussion in the results, no conclusion section. This needs to be rectified with careful attention as it makes the manuscript difficult to follow.
⇒ We changed the arrangement of the manuscript to conventional form.

3. I am confused as to why you present so many methods for deriving elevation change from CryoSat-2 data – I think there are three – one of which was introduced in the results section (see note above). Is this necessary? And if you believe so, please be clear why it is that you are doing this and which results are linked to which method.
⇒ To clarify, we moved all of the method in Result section to Method section.

4. You introduce ICESat results in the results section but there is no mention of this in the methods sec- tion. You need to include a description of this data and the processing methodology in you methods section.
⇒ We added the processing of ICESat data.

5. Throughout your text you report numbers of surface lowering from altimetry but you do not report associated errors – this makes it difficult to discern whether or not the changes in rates that you report are significant or not. I suspect in some instances they might not be. This must be rectified.
⇒ It was corrected all.

6. Also regarding errors, you report in you methods that you output and error statistic for the CryoSat elevation change signals from the regression error but later, in figures, use uncertainty measures from standard deviations of measurements and also from the signals from (assumed) constantly changing regions. I think it would be better/clearer/less confusing to stick to just one measure of uncertainty for any given parameter.

⇒ We clarified where the errors come from. Please read Method section.

**Specific points (given by Page number and Line Number – e.g. P1 L1):**

P1 L11-12 – change "inferred from CryoSat-2 altimetry, indicating that" to " inferred from surface height changes using CryoSat-2 altimetry and indicate that. . ."

⇒ It was corrected (P1 L12 -13)

P1 L13 – change "The orientation of the drainage network. . ." to 'The structure of the subglacial drainage network. . ."

⇒ We changed it as "The subglacial drainage network structure". (P1 L13)

P1 Introduction – general comment: I think that in the introduction you need to explain more about the previous work done into understanding the stagnation of the KIS. There is more than one theory for why this ice stream stagnated, and so I think that you need to acknowledge this.

⇒ It was corrected (P2 L1-5).

P1 L20 – this is a bit of an empty statement – explain why or to what extent the stagnation is significant for the mass balance of the WAIS – i.e. there is a mass gain.

⇒ We changed Introduction section so that reflect you and Reviewer2's comments (Your recommendation is reflected in P1 L26-27)

P2 L5 – you outline the limitations of the ICESat data, but what about the limitations of interferometry (which you mention in Line 3) for detecting SGLs ?

⇒ We added that contents. (P2 L15-16)

P2 L5 is this lake in the Northern Corner highlighted in any figure? If not, I would do so and include a reference here.

⇒ We refer a figure of Fried et al. (2014). (P2 L9-11)

P2 L10 – it would be helpful if you clearly stated the aims and objectives of your study here.

⇒ The last paragraph of Introduction section is changed.  (P2 L20-24)

P2 L 29 – why not just use a simple threshold at this stage?

⇒ The variances of Cryosat-2 elevations are too variety in this area and, moreover, the uncertainty of Cryosat-2 elevations is dependent on the surface slope. Therefore, the determination of a simple threshold over wide area for data editing is so difficult.

P3 L3 – Why not just use a smaller grid cell size? It is likely that there would be sufficient data within the grid cells at ~2 km to provide a robust solution.

⇒ We already have tried smaller (or bigger) grid cells. But in the case of smaller grid size, the noisy signals around subglacial lake were increased. Since our data processing is conducted in the time window of 2-year, the size of 2 km by 2 km is too small to gather enough measurements for stable linear fitting.

P3 L5 – regression of what? Do you mean the quadratic fit?

⇒ It was corrected. (P3 L17)

P3 L6 – is your selection of lake areas subjective or quantifiable? You need to explicitly state what criteria you use to determine if an area is lake or not.

⇒ It was corrected. (P4 L7-8)

P3 L9 -10 – these are results not methods

⇒ We corrected these sentences. (P2 L20-24)

P3 L10 – I would be interested to know if there are any potential lakes where you don't see any.

⇒ We also could find unknown subglacial lake in Whillans Ice Stream region. But we didn't stated in this paper because this study is only focused on KIS region.

P3 L11 – I don't understand why you have also use the plane fit method if this provides more detailed information on the boundaries. Also, I don't really understand how this can provide more detailed information on the boundaries, because the same data should be included in both methods.

⇒ The plain fit method was conducted by 5*5km grid cell overlapped 1 km interval and we just utilized it to finding candidates of subglacial lake. We could not use it to derive time series or lake boundary of the lake region, because it shows very low temporal and spatial resolutions (2-year window/ 5*5km grid cell). After we identify the possible existence of subglacial lake, we figured out the specific information of lake by a method of DEM difference with finer resolution (100 m). The structure of Data&Method section is changed.

P3 L12 state the resolution of these DEMs. Also what filtering criteria, if any, do you use to remove unreliable height estimates.

⇒ The resolution of DEM (100 m) was added (P4 L2). We applied a 3-sigma filter as mentioned in manuscript. In addition, the semivariogram modeling in kriging method includes nugget effect so that generates a smooth surface geostatistically.

P3 L18 – why do you think this is? Provide a physical explanation.

⇒ I think we need a more sophisticated statistical approach to the Cryosat-2 measurements, the ice surface morphology, and etc, but beyond the scope of this study. Therefore, a criterion for boundary selection was empirically chosen by comparison with ICESat measurement.

P3 L18-19 – lake areas should be in results.

⇒ It was corrected. (P5 L14-15)

P3 L19-20 – more detail needed for the methodology of deriving the hydraulic potential – at least a reference to the paper(s) from which you derived the method.
⇒ It was added in Method section. (P4 L34 – P5 L8)

P3 L25-29 – this is methods.
⇒ It was moved to Method section (P4 L10-20)

P3 L27 – state the magnitude and standard deviation of the trend removed? How much spatial variability was there in this trend?
⇒ Now more specific method is presented in Method section. (P4 L10-20)

P3 L 32- include uncertainty – this needs to be done for all your results (numbers) reported in the text.
⇒ It was corrected all.

P4 L 4 "descend completely yet" – replace with " returned to the previous level by the end of the study period."
⇒ It was corrected. (P5 L25-26)

P4 L7 – L17 – this is all discussion
⇒ It was moved. (P7 L12-22)

P4 L 20 – what are the values of head differences when the lakes are fully filled? Include numbers.
⇒ It was added. (P6 L3)

P4 L28 – you need to include your methods for using the LRM data in your methods section.
⇒ It was added. (P3 L5-11)

P4 L 35 replace "present" with 2015.
⇒ It was corrected (P6 L15)

P5 L 9 significant effect on what?
⇒ Reviewer 2 also point out that this sentence should be in Conclusion. We deleted this sentence in this section.

P5 L 9 – what exactly do you mean by the estuary? What characteristics does it have? In Figure 6 you show the area in a box – on what basis have you defined this area?
⇒ We added more specific statement and discussion in P8 L17~ P9 L3. In this area, the hydraulic potential lows are widening and flattening and the persistent elevation lowering are observed. Therefore, we think this area is an estuary where the subglacial flow enters into the ice shelf cavity.

P5 L11 – where has this ICESat data come from? Again you need to include this in your methods.

⇒ It was added (P4 L22).

P5 L 12 – How have you accounted for mission offsets in the elevations measured by the two missions? If you haven't, I don't think that you can meaningfully present the two as a 'continuous' time series on the same plot. Suggest you either show on separate plots, or show one plot of elevation changes (dhdt).

⇒ It was mentioned in Figure 6. The offsets of Crysat-2 elevations compared to the ICESat elevations due to mostly the penetration of radio wave into surface snow pack were estimated through the comparison between the ICESat and Cryosat-2 elevations at nearby stationary ice. The estimated offset (1.03 m) was subtracted from the ICESat elevation change time series for a continuous time series. We added it in the manuscript once again (P6 L26-27).

P6 L1 – explain why the feature is too concave for CryoSat-2 to detect.

⇒ It was added. (P7 L2)

P6 L8 – Change "If this method is applicable to the KIS, the strong. . ." to " If this method is applicable to the KIS, it is probable that the strong. . .."

⇒ This sentence was changed. (P9 L1)

P 7 – what about the conclusion section?

⇒ We changed the structure of this paper.

Figure 1 – I presume the yellow line is the Grounding Line? Include in figure caption.

⇒ It was included.

Figure 2 – There are two regions of surface lowering and 1 region of surface increase in the north of this figure? Why aren't these identified as lakes?

⇒ The two of the signals you pointed have large uncertainties. The small lowering signal between of them has low uncertainty, so it is another candidate of subglacial lake. However, it is ambiguous a bit because the elevation rate change of this region rapidly oscillate and also the location of this signal is slightly changes. The signal from this is distinct from the signals from KT2 and KT3, so it was excluded from our study.

Figure 3 – why not show continuous fields of the ice surface elevations and elevation anomalies?

⇒ In order to consider the locality of surface elevation, we made the local DEMs centered on each lakes using the kriging method based on local semi-variograms estimated in each areas.

Figure 4: why do you show the standard deviations here and not the uncertainty on the elevation change measurements that you derived in the processing?

⇒ We wanted to show the measurement error, not a regression error in elevation change time series. The measurement error from radar altimeter has been suggested in Wingham et al.,(2006b) and we chose their method. (P4 L16-19)

---

## Author Comment (AC2) · 14 Aug 2016

**Response to Anonymous Referee #2**

**We really appreciate for taking the time to review our paper and the comments valuable to improve our paper greatly. We address the comments in the order of the review as bellows. We noted the pages and line numbers indicating the corrections in revised manuscript corresponding to the comments. In addition, we would like to apologize a mistake in the analysis of the thinning in the estuary region. Owing to the referees' comments, we carefully validated the elevation time series from Cryosat-2 especially in the estuary region and found that the time series indicating the acceleration of thinning since 2013 were contaminated by the background elevation change (e.g. change of surface mass balance), because we could identify such changes in elevation on nearby stationary ice (e.g. even on the Siple Dome). We removed the background elevation changes around the estuary (modified Figure 6) and the statements related to the acceleration of thinning by subglacial flood in 2013. However, we believe it does not affect the main suggestion and conclusion of this paper.**

General Points:
1) Organization/structure – This paper at times lacks a clear organizational structure with introductions, methods, results, interpretation/discussion, and conclusions. I think that following this traditional structure could be useful in this case, as many different methods are being used and it is often difficult for the reader to determine the reason for selection of the method or analysis technique applied.
⇒ We changed the structure of the manuscript.

2) Methods and uncertainty – ICESat laser altimetry, surface elevation, and bed elevation data are used throughout the paper with clear explanation or citation. A thorough methodology and uncertainty analysis should be added to the manuscript. Regression error and standard deviation of residuals over non-changing portions are both used as measures of uncertainty. What are the reasons for including both uncertainty methods?
⇒ We modified the method section. The standard deviation of residuals over non-changing portions is to show the uncertainty of elevation change 'time series' as Figure 2 in Wingham et al. (2006b). For the uncertainty of elevation change 'rate', we used linear regression error. It is included now.

3) Labeling – Labeling in figures is often inconsistent. Sometimes panels are labeled; other times they are not. I'd prefer to err on the side of caution and label. Similarly, features in the figures are also often unlabeled or annotated. For example, grounding lines are not labeled or cited, but are plotted in the figures. Lakes outlines are also not consistently labeled, so it's unclear to me sometimes whether they are from Smith et al. (2009) or Wright and Siegert (2012) or this study.
⇒ We added labels in Figure 7. We also added annotations in Figure 1.

Scientific Points:
1) ICESat – ICESat laser altimetry data are extensively used but are not discussed. This is

especially crucial as these data are quantitatively compared to CryoSat-2 data, and the amplitudes of the elevation change derived are similar to the uncertainty of these methods. More discussion of the ICESat data and comparison between the ICESat and CryoSat-2 data would be useful to ascertain the significance of the signals analyzed.

⇒ The explanation of ICESat data was included.

2) Hydropotential – It is not clear what reference datum is used. It also seems as if two different datums are used as the hydropotential values differ significantly in the two figures shown. Finally, I assume that glaciostatic hydropotential (subglacial water pressure equal to overburden pressure) is calculated, but this is never stated.

⇒ We added about the calculation of hydropotential. You might feel confuse about the hydropotential in Figure 3. In this figure, the mean of hydropotential was removed for better visualization as stated in the caption.

3) subglacial lake detection algorithm – I would expect a more precise outline of the steps used, similar to the algorithm presented in Smith et al. (2009). For example, it's unclear to me what "visually inspect" means.

⇒ We modified the method section for clearer description.

4) CryoSat-2 data – Several methods are presented for analysis of the elevation data (quadrature curvature surface fitting, differences from reference DEMs, etc.). It's a little difficult to determine the exact sequence of methods being used and why. A more stepwise description and possibly flowchart figure would add clarity.

⇒ We changed the structure of method.

Style/Lanuage/Grammar comments

1) Hyphenation use is inconsistent – For example, "repeat track method" vs. topographyfree elevation" and "sub-glacial" vs. "subglacial". I tend towards hyphenation, i.e. repeat track and topography-free, but use should be consistent.

⇒ It was corrected all.

2) Capitalization use is inconsistent – For example, Antarctic ice sheet vs. Siple Coast Ice Streams. I would tend to lean towards Siple Coast ice streams and Antarctic ice sheet, but capitalization should be consistent regardless of convention.

⇒ It was corrected all.

**Specific Comments**

P1 L11: "We have identified two previously unknown active subglacial lakes…"

⇒ It was corrected (P1 L11)

P1 L12: Rapid fill-drain events do not necessarily indicate lake connectivity via a drainage network though they do indicate a subglacial drainage network exists.

⇒ A streamline from the regional hydropotential also indicates the connectivity lakes (P1 L13-14)

P1 L13: "lakes area" seems redundant. Perhaps just "lakes."
⇒ It was corrected (P1 L14)

P1 L14–16: This sentence should be rewritten to highlight the evidence that links subglacial lake drainages to the acceleration of thinning.
⇒ It was deleted.

P1 L15: "subglacial lakes".
⇒ We are sorry that we couldn't understand what this comment exactly means.

P1 L16–17: It seems unlikely to me that this conclusion can be justified from data presented here, which is too short a time series to suggest regions for the shutdown. It could be consistent with such a shutdown mechanism. This sentence should be reworded accordingly.
⇒ It was corrected (P1 L18-21)

P1 L16: Figure 2a does not seem to indicate rapid thinning. If anything, thickening seems the dominant signal. Perhaps the background elevation-change rate has been removed? Hopefully stated later.
⇒ All of the contents about rapid thinning is withdrawn now.

P1 L17: "sub-glacial" to "subglacial".
⇒ We choose "subglacial", and all of inconsistent uses are corrected.

P1 L19: Change "rapid ice flow" to "streaming ice flow".
⇒ It was corrected (P1 L25)

P1 L20–21: This sentence reads awkwardly. This paragraph could perhaps be restructured to facilitate reading. I would suggest the following general order:
1) Introduce ice-stream stagnation/reactivation cycles and their effects on ice-sheet mass balance.
2) Note that KIS stagnated ~160 years ago.
3) Note that changes in basal hydrology (rather via water piracy or change in dominant subglacial hydrology system structure) are likely a cause of this stagnation.
⇒ The arrangement of first paragraph in Introduction is changed (P1 L23~ P2 L5)

P1 L21: Change "indicated" to "posited", "hypothesized", "suggested", or similar to indicate hypothetical nature of this conclusion (even though it is likely correct).
⇒ It was corrected (P1 L28)

P1 L21: Change "Siple-coast" to "Siple Coast".
⇒ It was corrected (P1 L28)

P1 L22: Change "Ice Stream" to "ice streams".
⇒ It made the capitalization consistent (P1 L28)

P1 L22: Delete "of these cycles".

⇒ It was deleted.

P1 L24: Change "long term" to "long-term".

⇒ It was corrected (P1 L31)

P1 L24–25: Presumably it is associated with changes in basal-melt rate and upstream subglacial water supply, but how are these related? Are the authors suggesting a Tulaczyk et al. (2000) thermodynamic till mechanism or purely water piracy?

⇒ It was corrected (P1 L31-32)

P1 L25: "Therefore, it has been suggested…"

⇒ We rearranged the introduction section and this expression was deleted.

P1 L28: Change "predict its dynamics" to "understand the ice dynamics" or similar.

⇒ This expression was deleted.

P1 L30: Change "while" to "although".

⇒ It was corrected (P2 L2)

P2 L1: Delete "contributing the basal hydrology".

⇒ It was deleted.

P2 L3–4: "the hydrological connections between adjacent lakes".

⇒ It was corrected (P2 L8-9)

P2 L4: "by the sparse coverage of the ground tracks".

⇒ It was corrected (P2 L13-14)

P2 L5–7: Is this a supposition of the authors or are there other studies that posit this?

⇒ I was corrected. This supposition is posited in Fried et al., (2014) (P2 L9-11)

P2 L11–16: This paragraph mixes methods and introductory materials.

⇒ We changed this paragraph so that only contain an introductory material. (P2 L18-24)

P2 L11–12: Inconsistent capitalization: Antarctic Ice Sheet here vs. Antarctic ice sheet previously.

⇒ We use "Antarctic Ice Sheet" (P2 L18-19).

P2 L13: "two previously unknown subglacial lakes"

⇒ It was corrected (P2 L21)

P2 L14: What sort of activity? Downstream lakes fill as the upstream lakes drain?

⇒ We changed this sentence. (P2 L22-23)

P2 L15: Change "evidences" to "evidence".

$\Rightarrow$ It was removed.

P2 L13–15: This sentence is awfully general. Perhaps change to something more specific to the results presented in this manuscript.
$\Rightarrow$ We changed this paragraph. (P2 L18-24)

P2 L17: Seems like there should be a more extensive methods section.
$\Rightarrow$ We reconstructed the data and methodology section (P2 L25 – P5 L8)

P2 L21: "geophysical" should probably be "geographic".
$\Rightarrow$ We changed this sentence. (P2 L32 – P3 L1)

P2 L23: Which version of the L2 product? If I remember correctly, there are multiple processing baselines.
$\Rightarrow$ As you noted, we used both of baseline B and baseline C products. Our results shown in manuscript are already corrected the bias between those products. We added this statement in P3 L2-4.

P2 L24: What does "interior of ice" mean? Ice-sheet interior? Why the range of uncertainty?
$\Rightarrow$ It was corrected (P2 L31). Wang et al. (2015) suggests that the uncertainty is proportional to the slope of surface (P2 L30-32)

P2 L24–26: I am confused by this sentence. There seem to be multiple numbers for uncertainty and their dependence on physical values varies. A more complete uncertainty approach is probably needed, or at least a more complete discussion of the approach adopted here.
$\Rightarrow$ Wang et al. (2015) computed the uncertainty of Cryosat with respect to ICESat measurement, and the uncertainties depended on the slope of ice surface. The uncertainty of Cryosat measurement on KT region is also estimated in this study, and presented in Figure 4, 5 and revised method section.

P2 L27–P3 L10: I think a step-by-step description and/or processing flow chart is needed here. As is currently written, the description of the processing steps is somewhat unclear.
$\Rightarrow$ The methodology section is changed (Section 2.2 - 2.4).

P2 L28: What do height error flags signify?
$\Rightarrow$ This sentence was changed (P3 L1-2).

P2 L30: Does this method follow Helm et al. (2014) or is it distinct in some way from that processing?
$\Rightarrow$ Helm et al. (2014) is a reference of the DEM product used for 3-sigma filtering.

P3 L3–5: What does visually inspect actually mean? To be robust, I think quantitative metrics are required to determine reliability of subglacial lake detection. A protocol like that described by Smith et al. (2009) seems to be needed here.
$\Rightarrow$ We changed Data & Method section (Section 2.2 - 2.4).

P3 L5: What are the uncertainties estimated from the regression? It is hard to ascertain what the numbers actually are from Figure 2b.
⇒ It was added (P3 L19-20).

P3 L6: What is sufficiently lower? A more rigorously quantified uncertainty analysis is needed.
⇒ We added those values (P3 L28-29).

P3 L12: In various time windows? Different windows seem to represent different lengths of time? I would have thought an overlapping window scheme of a constant time length would be used? If these irregular windows can be easily justified, I'd like to know why.
⇒ We described in details (P3 L20). The detection of lake was performed using an overlapping window scheme of a constant time length (2 year). However, in order to highlight the elevation change of lake and clearly select the lake boundary, we use the time windows with different lengths for generating DEMs as described in section 2.3.

P3 L19: I assume this is glaciostatic hydropotential following Shreve (1972), but it would be nice to have this verified at least once.
⇒ We added it in Method section. (P4 L34 - P5 L8)

P3 L19–20: Why not say directly that the "lakes are located in hydropotential lows."
⇒ It was corrected and moved to result section (P5 L11-12)

P3 L24–25: This sentence does not seem necessary.
⇒ It was removed.

P3 L27–28: "The background temporal elevation changes outside the lake boundaries are removed to examine only the elevation changes associated with the SGLs activity." How was this done precisely?
⇒ We added it in Method section. (Section 2.4)

P4 L1: Are the elevation change numbers sufficiently robust to actually derive this balance flow rate? What is the associated uncertainty?
⇒ We added their uncertainties. (P5 L22)

P4 L1–4: This reverse sequential drainage seems odds compared to the sequential filling. If KT1 is supplying the other two lakes, why does it not continue filling if the water is no longer flowing to the other lakes? Does it go someplace else? Or has the inflow rate just slowed? In Figure 4, the fluctuations seem large relative to the uncertainty. Is the uncertainty here truly representative of that in these data? Can these high-amplitude fluctuations be interpreted?
⇒ One possible scenario has been stated in P7 L12-22. The high-amplitude fluctuations observed when the lakes were filled is probably due to spatially irregular sampling of the elevation change pattern like a dome within the lake boundary. This speculation was added (P5 L26-28).

P4 L5–15: Much of this seems speculative and just descriptive of the results of other studies.

Are the flux rates from these lakes sufficient to maintain a connected network of Röthlisberger channels as the authors seem to suggest here? Alternate theories also seem possible. The simplest explanation may be that these lakes, separated by ~100 km, simply are not hydraulically connected and fill and drain separately. I am hesitant to put much faith in hydropotential maps on this level and much of the inflow from KT3 appears to come from a separate upstream hydraulic catchment anyway. If the authors are suggesting that a channelized hydrology system can be maintained against creep closure via this water flux, I'd like to see some calculations and type of channel suggested (R- or N-channel, canal network, etc.). I note that the one real-time observation of transient water flow under an ice stream (i.e., not a subglacial lake filling or draining, but subglacial water in motion; see Winberry et al. (2009)) indicates that the channels are probably not maintained for more than a few weeks, not the many months suggested here.

⇒ We added more discussion in P7 L22 – P8 L11. In the assumption of the R-channel, the energy analysis similar to Wingham et al. (2006b) shows the energy released by the subglacial flood between KT1 and KT2 is enough to maintain the semi-circular conduit against creep closure. However, the R-channel theory may be not adequate in the environment of Antarctica, especially Siple Coast. Therefore, we referred a recent study investigating the possibility of a canal flow in the Whillans/Mercer Ice Stream.

P4 L19: It is hard to verify the hydraulic head difference cited here from the figures. Looking glibly at Figure 4, the hydraulic head changes seem like they should be somewhat less than the numbers cited here. However, it's difficult to tell if the authors are including flow focusing or some other effect in their calculation. More details are needed.
⇒ We clarified it in Figure 4 and in P6 L2-4

P4 L23: "role of the hydraulic barrier" or "role of hydraulic barriers".
⇒ It was corrected (P7 L18-19)

P4 L25–28: This sentence could be split into two or a comma should be added to separate clauses: "…KIS area, but the…".
⇒ It was corrected (P6 L8)

P4 L28: How are the LRM products used? This should be added to the methodology, or at least there should be a citation if following an established method. LRM mode differs significantly from SARin, and thus differences in uncertainty, reliability, resolution, etc., would be expected; clarification of these differences is needed.
⇒ We added the description about LRM products in Section 2.1 (P3 L5-11).

P4 L32 – P5 L1: This is more like the patter of draining I would expect. I wonder if an extended discussion of the difference between the upstream and downstream lakes would be useful.
⇒ A sentence was added in discussion section (P7 L19-21)

P5 L8–9: This sentence seems like a conclusion before the data/interpretation/discussion are presented.
⇒ It was removed.

P5 L12: Methods on how the ICESat and CryoSat-2 time series are combined are necessary. To do this properly, error and biases should be clearly presented.

⇒ It has been suggested in the caption of Figure 6. We added it in the manuscript once again (P6 L26-27)

P5 L15–17: Can you really assess when precisely KT1 stopped draining with the elevation amplitude presented in Smith et al. (2009). I would think the ICESat data would allow more precise determination of subglacial lake drainage timing. A drainage lasting this long seems unlikely compared to other subglacial lake drainages documented (c.f., Siegfried et al., 2016).

⇒ Although not shown in this paper, the ICESat elevations within KT1 from our processing are continuously and slowly lowering in a rate of ~ -0.3 m/yr in consistent with Smith et al. (2009). However, we cannot find a significant elevation change in the Cryosat-2 elevations from 2010 to 2012. We suppose the slow draining of KT1 observed by ICESat was stopped around 2009.  After the filling event in 2013, the KT1 seems to be slowly lowering similar to the observation during ICESat era but we need to monitor it for a few more years in order to figure out the long-term draining or its periodicity.

P5 L19–22: How does this estuary compare to the one documented on Whillans Ice Stream (see Horgan et al. (2013a,b) and Christianson et al. (2013)). A discussion of similarities and differences between the estuaries would be important, as progradational till deltas were directly observed in that estuary and it seems as the authors are suggesting a similar depositional structure here, with sheet (distributed) flow and channelized flow coexisting.

⇒ We added more discussions that you recommended. (P8 L26- P9 L3).

P5 L23–24: Enhanced lubrication isn't a necessary condition for this. Tensile forces must inherently exist at the junction of an ice stream and ice shelf (see Weertman (1974) and Schoof (2007) among others). Although additional basal lubrication could result in increased longitudinal stress. The timing of the lake drainage does seem nicely correlated with the increase thinning, but it could have but both events could have been triggered as a result of regional grounding line retreat, and thus the lake drainage could have been an effect and not a cause. Some discussion of the nuances and limitations of the data would be helpful, as well as connecting more directly to known background thinning rates and ice bottom/bedrock geometry.

⇒ This argument is deleted now.

P5 L30: These feature looks distinct from the channel described in Marsh et al. (2016), which does not have undulating topographic features in the along flow direction. Some discussion of why this might be would be useful.

⇒ The features of channel described in Marsh et al. (2016) is mainly from MOA image. I MOA image, we only can see the dark and flat features from KIS cavity and we think MOA is not proper to find out the specific morphology. Since the undulating feature is from Landsat imagery, we cannot simply compare the observation of Marsh et al. (2016) with our observation directly and discuss although we have some speculations.

P5 L33: I am suspicious of the 30 m/yr retreat rate. Grounding-line location was not well

known along the Siple Coast in the late 1980s. Is there no newer result?

⇒ Although not shown in this paper, the differencing of Landsat images (using Landsat 7 and Landsat 8 scenes) gives us the retreat rates of 30 - 50 m/yr along the grounding line of KIS in consistent with Thomas et al. (1988), so we referred it instead of including the result.

P6 L6: "to the oceans".

⇒ This sentence was changed (P8 L32-34).

P6: L4–8: There should probably be a citation to the buoyant meltwater plume circulation that drives this circulation in the channel – I'd suggest Jenkins (1991) and Jenkins (2011). Ruling out other possible channel creation mechanisms is probably needed too, i.e., highly variable bed topography, suture zones, etc.

⇒ We referred Jenkins (2011). Based on the result in Jenkins (2011), the steep ice base near the grounding line of KIS trunk estuary observed in BEDMAP2 may support the strong basal melting by meltwater plume. (P9 L4-6)

P6 L12–13: Is the "observed" channel in the same location as the modeled ones?

⇒ The location of observed channel is nearly similar with the modeled one in previous studies (see Figure 3 in Carter et al.,2012 and Figure 2 in Goeller et al., 2015).

P6 L15: Here and throughout the manuscript I wonder if the distributed rather than sheet flow might be what the authors are describing. Sheet flow implies a thin water film a few millimeters thick. Distributed flow would allow more generality.

⇒ We corrected them all that you recommended.

P6 L19–21: Much of this seems like conjecture. Other drainage systems besides a strictly channelized system could lead to relatively rapid connectivity between lakes. Small outburst floods with transient channelization (see Winberry et al. (2016)) seem more likely to me than a long-lived R-channel connecting lakes. The flow of water along paths down the hydropotential gradient does not necessarily imply channelized flow either.

⇒ We cannot decide the kind of flow in current observation. We corrected its expression so that it could be opened to other possibilities. (P7 L22-P8 L11)

P6 L30: Perhaps "Comparison of our results to Siegfried et al. (2016)" to avoid confusing the data presented here with those derived from studies of Whillans Ice Stream subglacial hydrology.

⇒ It was corrected (P9 L14-15)

P7 L5: "definitely" should perhaps be "definitively"?

⇒ It was corrected (P10 L2)

P7 L5–8: Perhaps reword as: "At present, our results cannot definitively determine whether KIS stagnation occurred via basal water channelization, water piracy, or some combination of these. Further studies of basal water flow in the KIS trunk would be necessary to make this

determination." I don't see the need for channelization or water piracy to be mutually exclusive.

⇒ It was changed by the similar sentence (P10 L2-4)

Figure 1: Smith et al. (2009) or other sources should be cited for location of previously known subglacial lakes. Grounding line (yellow) and appropriate citations should be given in the caption. What is the date of this grounding line? Citations for MEaSUREs velocity data and MOA imagery are similarly needed (I cannot tell which version of each was used). Some reference should be made to lake outlines shown for KT1, KT2, and KT3 (even if "as discussed in the text."). Some demarcation should be shown for Figure 5–7 (or shown sequentially if not marked here). This is especially crucial for Figure 5, as there are not good references to scale for that figure.

⇒ Figure 1: It was corrected that you recommended.

Figure 3: What datum is used in hydropotential calculations. Although the gradients are the amplitude I'd expect, the value of hydropotential itself seems unlikely to be slow. Theoretically hydropotential should go to 0 at the grounding line. Although some values of hydropotential may drop below 0 on grounded ice, this seems to be widespread. I suspect this is a result of the datum being used.

⇒ We are sorry for that we didn't state the mean of hydropotential is removed in figure (c). we added this statement.

Figure 6: Once again, hydropotential values seem off. They should go to near 0 near the grounding line. Even using a standard datum (WGS84 ellipsoid; EGM2008 geoid) would get relatively close. These seem too high. These values are also inconsistent from those shown in Figure 3. Grounding line (in yellow) should be noted in caption. How was the estuarine area shown in red determined? Why is there no uncertainty on the elevation change plot (d) when there are uncertainties on similar plots in other Figures? Label ICESat tracks shown in Figure b.

⇒ The color scale made you feel confused about it. Note that the color axis was cut at 500kPa although the hydropotential actually go to nearly zero around the grounding line, because we want to highlight the potential differences of KT2, KT3 and the estuary near grounding line. The hydropotential rapidly changes near the grounding line because the ice base is very steep. The BEDMAP2 uses GL04C geoid as a datum for elevations.

Figure 7: Though not particularly important, I am surprised gap filling did perform better for the November 2011 Landsat 7 scene. I assume the fit in the final panel is linear, but it would be good to note this. Uncertainty values in elevation-change panel would also be desirable. I suspect that labeling the panels (a–e) would be useful here too.

⇒ We added labels in this figure and also added that the linear fit is performed.

---

## Referee Report (RR1)

**General Comments:**

This manuscript uses satellite laser and radar altimetry to identify two previously unidentified subglacial lakes on Kamb Ice Stream, and discusses the implications of this discovery for ice dynamics.

This is my second time reviewing this manuscript and I found it much improved. The quantitative treatment of the CryoSat-2 data is significantly better, both in how it was explained and in how it was actually executed. My remaining comments are much more minor. My primary scientific issues are concerning interpretation, both the calculations concerning R-channel size and stability as well as the discussion of the existence of the channel in the subglacial estuary. If these areas can be improved, I believe this manuscript will become a nice addition to the Cryosphere.

I list more specific scientific, stylistic, and specific comments below.

**Scientific Issues:**

1. Uncertainty could still be clarified in some cases. I noted particular issues in the specific comments sections.

2. What were the length and duration of time windows tests? I am just concerned about spurious effects from irregular sampling and/or elimination of small elevation changes?

3. The discussion of R-channels under the grounded ice stream is a confusing. The first part is fine, but I am not actual sure the pressure differential supplied by the lake is sufficient to sustain a channel. I also agree with the authors that other mechanisms are likely active under ice sheets.

4. The arguments for/against a subglacial channel in the estuary are conflicting. I think the arguments for a subglacial are more convincing and suggest the authors revisit that section to make it consistent reasoning throughout.

**Style/Organizational Comments:**

1. When describing the CryoSat products, it would probably be best to start with a description of the different modes and then proceed to discuss them. Right now the text jumps backwards and forwards too much between these, which I found confusing as a reader. Probably best to start with geographic coverage, then discuss precise imaging characteristics, and then finally to proceed to uncertainty. This was better done for LRM than for SARin mode.

2. The authors occasionally switch from present to past tense. It's best to just use one convention. I'm not particularly partial to one or the other, as long as the use is consistent.

**Specific Comments:**

**Page 1**

12: What is the timescale for rapid? Over a few months?

13: Add "…subglacial drainage network…"

14: I worry about using "clearly links the lakes" here when it was not directly observed. Replace with "likely"?

21: What does "subdued" mean? Fewer lakes or longer fill–drain cycles?

**Page 2**

10: change "margin" to "marginal"

12–13: Does work by either Le Brocq et al. (2013) or Alley et al. (2016) indicate the existence of the channel? If I remember correctly both do indicate a channel and it might be good to cite some observational evidence too.

21: Something is missing in this sentence. Perhaps "report the existence of" or "report the presence of"

31–32: Does the range in uncertainty you quote here include the slope error? If not, how much does that increase local uncertainty?

**Page 3**

10–11: What are uncertainties of LRM L2 products?

15: What sorts of errors does these height error flags indicate?

16–17: Perhaps state what data were used to make the DEM product?

24–25: What are the errors precisely? RMS or residuals to linear fit, standard error, or something else?

28–29: Use "surface elevation change rates" instead of "large change rates" for clarity?

**Page 4**

4: How many time windows did you test? Was there a standard procedure for this?

23–24: Citation for these figures?

25–26: Citation for ICESat filtering algorithm applied here?

**Page 5**

20: Is there unit issue here? Perhaps volume variations instead of elevation variations?

26-27: It should be mentioned that SARin mode samples high points and can thus bias observations when this mode is initially described back in section 2.1. That would make understanding the origin of the biases mentioned here easier.

**Page 6**

1: Refer to the section where this is discussed in parenthesis.

6–7: Reverse the clause ordering, so that "According to…" begins the sentence so that the reader knows this is a based on a previous modeling result.

22: Perhaps note that this is similar to the estuary documented by Horgan et al. (2013).

26: Is this radar penetration into the snow similar to what is found by other studies? Or is this purely an empirical conclusion?

30: Again, possibly cite Le Brocq et al. (2013) and Alley et al. (2016) papers if they also indicate a subglacial channel in this location.

32: I'm not sure retreat rate is the right word here. The feature is extending inland but perhaps not retreating? In any case, some precision with language would be useful here so that the reader knows that you are not referring to grounding-line retreat.

**Page 7**

4: "Posit" instead of "suppose".

11: Capitalize "lake" in "lake Conway" and "lake Mercer" for consistency.

15–21: Some care may be needed with language to stress that these are hypothesized linkages and relationships between lakes.

17–18: Reword: "The water drained from KT2 that passes directly through KT3…"

19–21: Reword: "In comparison to the behaviors of K1 and K34, which show more typical connectivity for subglacial lakes…"

**Page 8**

1–2: This discussion is a bit muddled. The first part is reasonable clear. However, the discussion of effective pressure is not. The perturbation in the effective pressure necessary to generate the tunnel is the important quantity, so that when the elevation is high sufficient pressure is present whereas when it is not present, the effective pressure is too low to maintain the channel. Furthermore, a more complex consideration of effective pressure over a long flow path is probably needed if considering more than a local effect right at the lake boundary.

10–11: Inflow into WIS is transient too, so I don't see an issue with this. There's no reason to suppose that an N-channel into the sediment cannot be closed and opened by repeated lake drainage, likely in approximately the same location (as dictated by the basal hydropotential) but not precisely the same flowpath.

13–15: I'm not particularly convinced by this, especially as I think others have observed channel-like features in MODIS imagery here.

29–30: I like this hypothesis better than suggesting only channelized flow.

33–34: This is really just an application of the Jenkins (1991) model and that should be cited too.

**Page 9**

4–6: I think this reasoning for the existence of a large subglacial channel is more consistent with the observations, but it contradicts earlier statements. I suggest revising the wording in that section.

20: Bridging "stresses" instead of "forces".

**Page 10**

2–3: Perhaps say that both mechanisms could be active and are not necessarily mutually exclusive. Less water overall, but also more channelized where there is water such that little lubrication can be supplied by the subglacial water.

Figure 3: In caption, state that red rectangles are at the center of the hydropotential lows.

Figure 4 (and 5b and 5c): Could volume change be plotted on the right y-axis as the shapes of the curve are identical?

**References**

Alley, K. E., T. A. Scambos, M. R. Siegfried, and H. A. Fricker (2016), Impacts of warm water on Antarctic ice shelf stability through basal channel formation, Nature Geosci. 9, 290–293, doi: 10.1039/ngeo2675.

Horgan, H. J., R. B. Alley, K. Christianson, R. W. Jacobel, S. Anandakrishnan, A. Muto, L. H. Beem, and M. R. Siegfried (2013), Estuaries beneath ice sheets, Geology, 1159–1162, doi:10.1130/G34654.1.

Jenkins, A. (1991), A One-Dimensional Model of Ice Shelf–Ocean Interaction, Journal of Geophysical Research 96(C11), 20671–20677.

Le Brocq, A. M., N. Ross, J. A. Griggs, R. G. Bingham, H. F. J. Corr, F. Ferraccioli, A. Jenkins, T. A. Jordan, A. J. Payne, D. M. Rippin, and M. J. Siegert (2013), Evidence from ice shelves for channelized meltwater flow beneath the Antarctic Ice Sheet, Nature Geosci. 6, 945–948, doi: 10.1038/ngeo1977.

---

## Author Response (AR2)

**Dear Editor,**

We really appreciate the careful review and all the constructive suggestions and comments. We have revised our manuscript in reference to the referee's and editor's comments. The answers are as bellows.

**Responses to editor's comments**

Figure 1: Label white, black and red rectangles as "Fig. 2", "Fig. 5" and "Fig. 6" so that the figure is more self-explanatory.
→ It was added(P14).

Figure 3:
- The caption is confusing. The first sentence says elevations and elevation anomalies but three panels show these two and hydraulic potential anomaly. Is it better to say "Ice-surface elevations and subglacial hydraulic potential around the three subglacial lakes KT1, KT2, and KT3 overlaid on the MOA image."?
→ It was changed that you recommended. (P15)

- Both (b) and (c) show local anomalies or relative values. It's easier to read if both are said "XXX anomaly" (elevation anomaly and hydraulic potential anomaly).
→ It was changed(P15).

- "The red square indicates the location of the hydro-potential lows and the white square indicates the predicted outlets"?
→ It was changed(P15).

Figure 4: the transparent color shows one standard deviation of elevation measurements on the stationary ice adjacent to each lake. It's a fact. I think that the authors judge that one standard deviation is a good measure of uncertainty but it is interpretation. Please separate the explanation to the fact indicated in the figure and its interpretation.
→ We moved that sentence to the result section (P5 L22-24)

Figure 5: At the end of panel a's caption, add "See Fig. 1 for the lake K8's location" or such.
→ It was added(P17).

Figure 6:
- Typo. 3 retangles should be "rectangles".
→ It was fixed(P18).
- No gray rectangle (the last sentence in the caption).
→ The last sentence was removed.

English language
→ We improved the quality of English language, grammar, readability, and word choice by the helps of a native co-author and a commercial English editing company.

**Responses to reviewer's comments**

General Comments:
This manuscript uses satellite laser and radar altimetry to identify two previously unidentified subglacial lakes on Kamb Ice Stream, and discusses the implications of this discovery for ice dynamics.
This is my second time reviewing this manuscript and I found it much improved. The quantitative treatment of the CryoSat-2 data is significantly better, both in how it was explained and in how it was actually executed. My remaining comments are much more minor. My primary scientific issues are concerning interpretation, both the calculations concerning Rchannel size and stability as well as the discussion of the existence of the channel in the subglacial estuary. If these areas can be improved, I believe this manuscript will become a nice addition to the Cryosphere.
I list more specific scientific, stylistic, and specific comments below.

Scientific Issues:
1. Uncertainty could still be clarified in some cases. I noted particular issues in the specific comments sections.
→ It was modified.

2. What were the length and duration of time windows tests? I am just concerned about spurious effects from irregular sampling and/or elimination of small elevation changes?
→ A year and two year time windows were mainly tested. As you expected, we could not apply this method in the rugged terrain (higher uncertainty region in height derivation) because it can also remove the actual elevation change signal in the process of filtering. However, at least in the Siple Coast Ice Stream we could successively derive the elevation change over active subglacial lake, because the noisy data was less than other region due to relatively flat surface.

3. The discussion of R-channels under the grounded ice stream is a confusing. The first part is fine, but I am not actual sure the pressure differential supplied by the lake is sufficient to sustain a channel. I also agree with the authors that other mechanisms are likely active under ice sheets.
→ Preventing the confusing, a sentence was modified. See the answer for the comments about the 1-2 line of page 8.

4. The arguments for/against a subglacial channel in the estuary are conflicting. I think the arguments for a subglacial are more convincing and suggest the authors revisit that section to make it consistent reasoning throughout.
→ Perhaps it is because of a paragraph address about distributed flow. We actually suggest a small part of channelized flow is converted to distributed basal flow. Some words were slightly changed so that support both of channelized and distributed flow.

Style/Organizational Comments:

1. When describing the CryoSat products, it would probably be best to start with a description of the different modes and then proceed to discuss them. Right now the text jumps backwards and forwards too much between these, which I found confusing as a reader. Probably best to start with geographic coverage, then discuss precise imaging characteristics, and then finally to proceed to uncertainty. This was better done for LRM than for SARin mode.
→ We re-arranged that paragraph (P2 L29-P3 L7).

2. The authors occasionally switch from present to past tense. It's best to just use one

convention. I'm not particularly partial to one or the other, as long as the use is consistent.
→ It was corrected.

Specific Comments:

12: What is the timescale for rapid? Over a few months?
→ It was added (L12).

13: Add "…subglacial drainage network…"
→ It was added (L13).

14: I worry about using "clearly links the lakes" here when it was not directly observed.
Replace with "likely"?
→ It was corrected (L14).

21: What does "subdued" mean? Fewer lakes or longer fill–drain cycles?
→ It was changed to "the degree of KIS trunk subglacial lake activitiy is relatively weaker than those of the upstream lakes" (L20-21)

10: change "margin" to "marginal"
→ It was corrected (L11).

12–13: Does work by either Le Brocq et al. (2013) or Alley et al. (2016) indicate the existence of the channel? If I remember correctly both do indicate a channel and it might be good to cite some observational evidence too.
→ We added Alley et al. (2016). Le Brocq et al. (2013) did not address about KIS (L15).

21: Something is missing in this sentence. Perhaps "report the existence of" or "report the presence of"
→ It was corrected (L23).

31–32: Does the range in uncertainty you quote here include the slope error? If not, how much does that increase local uncertainty?
→ The range includes the slope error.

10–11: What are uncertainties of LRM L2 products?
→ The uncertainty of LRM L2 product has been well assessed over ocean, but we couldn't find any researches which fully assessed the accuracy of LRM product over the interior of the Antarctica.

15: What sorts of errors does these height error flags indicate?
→ The height error flag of Cryosat-2 L2 products indicate whether the height derivation by the retracker of L2 product success or not.

16–17: Perhaps state what data were used to make the DEM product?
→ It was added. Helm et al. (2014) used the Cryosat-2 data to make the DEM (L20).

24–25: What are the errors precisely? RMS or residuals to linear fit, standard error, or something else?
→ It was changed to "regression error" (L30).

28–29: Use "surface elevation change rates" instead of "large change rates" for clarity?
→It was corrected (L31).

4: How many time windows did you test? Was there a standard procedure for this?
→ We empirically set the time window by inspecting the initial elevation change time series (sampled from rectangular boundary around each subglacial lake). Then, we calculated more precise boundary and sampled elevation changes again.

23–24: Citation for these figures?
→It was corrected (L27-28).

25–26: Citation for ICESat filtering algorithm applied here?
→We cited Pritchard et al. (2012) (L29-30).

20: Is there unit issue here? Perhaps volume variations instead of elevation variations?
→ It was corrected (L26).

26-27: It should be mentioned that SARin mode samples high points and can thus bias observations when this mode is initially described back in section 2.1. That would make understanding the origin of the biases mentioned here easier.
→It was added in Section 2.1 (P3 L2-5).

1: Refer to the section where this is discussed in parenthesis.
→ It was added (L6).

6–7: Reverse the clause ordering, so that "According to…" begins the sentence so that the reader knows this is a based on a previous modeling result.
→ It was changed (L11-12).

22: Perhaps note that this is similar to the estuary documented by Horgan et al. (2013).
→ This similarity will be presented in the Section 4.

26: Is this radar penetration into the snow similar to what is found by other studies? or is this purely an empirical conclusion?
→ We cited Davis and Moore (1993), which have studied the volume scattering in the snow pack for radar altimetry.

30: Again, possibly cite Le Brocq et al. (2013) and Alley et al. (2016) papers if they also indicate a subglacial channel in this location.
→ We added Alley et al. (2016) (P7 L2).

32: I'm not sure retreat rate is the right word here. The feature is extending inland but perhaps not retreating? In any case, some precision with language would be useful here so that the reader knows that you are not referring to grounding-line retreat.
→ "retreat rate" was changed to "extending rate" (P7 L4) .

4: "Posit" instead of "suppose".
→ It was changed (L9).

11: Capitalize "lake" in "lake Conway" and "lake Mercer" for consistency.
→ It was changed (L17).

15–21: Some care may be needed with language to stress that these are hypothesized linkages and relationships between lakes.
→ It was changed (L21-26).

17–18: Reword: "The water drained from KT2 that passes directly through KT3…"
→ It was changed (L23).

19–21: Reword: "In comparison to the behaviors of K1 and K34, which show more typical connectivity for subglacial lakes…"
→ It was changed (L25-26).

1–2: This discussion is a bit muddled. The first part is reasonable clear. However, the discussion of effective pressure is not. The perturbation in the effective pressure necessary to generate the tunnel is the important quantity, so that when the elevation is high sufficient pressure is present whereas when it is not present, the effective pressure is too low to maintain the channel. Furthermore, a more complex consideration of effective pressure over a long flow path is probably needed if considering more than a local effect right at the lake boundary.
→ It was modified as "However, the increase in pressure at lake KT2 ($\rho_i g \Delta h$), where $g$ is gravitational acceleration and $\Delta h$ (= 1.7 m) is the elevation change of lake KT2, is ~17 kPa, which is much smaller than the effective pressure required for the creep close of tunnel (~700 kPa).". As commented, this simple analysis may be not adequate to explain the flow along the long flow path. We think further studies by us or other researchers are required as you mentioned. (L6-8)

10–11: Inflow into WIS is transient too, so I don't see an issue with this. There's no reason to suppose that an N-channel into the sediment cannot be closed and opened by repeated lake drainage, likely in approximately the same location (as dictated by the basal hydropotential) but not precisely the same flowpath.
→ It was removed because it was not a clear sentence as you commented.

13–15: I'm not particularly convinced by this, especially as I think others have observed channel-like features in MODIS imagery here.
→ We also agree with that channel-like feature also exist beneath the estuary region because the existence of sub-ice-shelf cavity supports it. We suggest here a small part of channelized flow is converted to distributed basal flow. We slightly changed the next sentence so that it supports both of flows. (L17-18)

29–30: I like this hypothesis better than suggesting only channelized flow.
→ The previous paragraph was partly changed.

33–34: This is really just an application of the Jenkins (1991) model and that should be cited too.
→ It was added (P9 L7).

4–6: I think this reasoning for the existence of a large subglacial channel is more consistent with the observations, but it contradicts earlier statements. I suggest revising the wording in that section.
→The previous paragraph was partly changed (P8 L17-18).

20: Bridging "stresses" instead of "forces".
→ It was changed (L27).

2–3: Perhaps say that both mechanisms could be active and are not necessarily mutually exclusive. Less water overall, but also more channelized where there is water such that little lubrication can be supplied by the subglacial water.
→ It was added (L11).

Figure 3: In caption, state that red rectangles are at the center of the hydropotential lows.
→ It was corrected (P15 L9).

Figure 4 (and 5b and 5c): Could volume change be plotted on the right y-axis as the shapes of the curve are identical?
→ Since the volume changes are scaled by the lake areas respectively, the right y-axis for volume change is not proper for this case.

Added references

[revised manuscript text omitted]

**(a) December 1991**

3 km

**(b) December 2001**

**(c) November 2011**

**(d) Dec. 2001 - Dec. 1991**

**(e) Nov. 2011- Dec. 2001**

221

112

0

-1.5 m/yr

**(f) ICESat Elevation**

dh/dt=-1.2⊥0.06 m/yr

ELEV. CHANGE (m)

YEAR

**Figure 7: Landsat images ((a) – (c)) and their difference images ((d) – (e)) over the region indicated by the blue box in Figure 6. The stripes in the image for November 2011 due to the failure of the scan line corrector (SLC) of Landsat-7 ETM+ are removed by a gap-filling method using additional images taken around the same time. The elevation change rates from the ICESat measurements are overlaid on the difference image (e). Panel (f) shows the temporal elevation change in the topographic sink, measured along the 221 ICESat track.**

---

## Author Response (AR3)

**Dear Editor,**

We have revised our manuscript in reference to an editor's comment for final production. We included a sentence in P3 L26-L28: "A test in higher temporal resolution, i.e. using the time window of one-year, severely increases the uncertainties of elevation change rate anomalies due to the sparsity of data sample in the $5 \times 5$ km regions, though it is not shown here.". We really appreciate the careful review and all the constructive suggestions and comments.

Sincerely,

Choon Ki Lee